# Src-dependent phosphorylation of μ-opioid receptor at Tyr[336] modulates opiate withdrawal

Lei Zhang[1,†] iD, Cherkaouia Kibaly[1,†,*] iD, Yu-Jun Wang[2,†] iD, Chi Xu[1] iD, Kyu Young Song[1] iD, Patrick W McGarrah[1], Horace H Loh[1], Jing-Gen Liu[2,**] iD & Ping-Yee Law[1,***] iD

## Abstract

Opiate withdrawal/negative reinforcement has been implicated as one of the mechanisms for the progression from impulsive to compulsive drug use. Increase in the intracellular cAMP level and protein kinase A (PKA) activities within the neurocircuitry of addiction has been a leading hypothesis for opiate addiction. This increase requires the phosphorylation of μ-opioid receptor (MOR) at Tyr[336] by Src after prolonged opiate treatment *in vitro*. Here, we report that the Src-mediated MOR phosphorylation at Tyr[336] is a prerequisite for opiate withdrawal in mice. We observed the recruitment of Src in the vicinity of MOR and an increase in phosphorylated Tyr[336] (pY336) levels during naloxone-precipitated withdrawal. The intracerebroventricular or stereotaxic injection of a Src inhibitor (AZD0530), or Src shRNA viruses attenuated pY336 levels, and several somatic withdrawal signs. This was also observed in Fyn[−/−] mice. The stereotaxic injection of wild-type MOR, but not mutant (Y336F) MOR, lentiviruses into the locus coeruleus of MOR[−/−] mice restored somatic withdrawal jumping. Regulating pY336 levels during withdrawal might be a future target for drug development to prevent opiate addictive behaviors.

**Keywords** lentivirus injection; locus coeruleus; naloxone-precipitated opiate withdrawal; opiate addiction; Src-mediated phosphorylation of MOR at Tyr[336]

**Subject Category** Neuroscience

## Introduction

Cravings for opioids have been linked to the initial rewards associated with the drugs as well as to withdrawal and the adverse motivational state (Koob, 2009). The neurocircuitry involved in the three stages of the addiction cycle has been mapped, with the extended amygdala as the focal point in the withdrawal/negative affect stage (Stinus *et al*, 1990; Nestler, 2004; Koob & Volkow, 2010). The extended amygdala is composed of the central nucleus of the amygdala, the bed nucleus of the stria terminalis (BNST), and the medial (shell) subregion of the nucleus accumbens (NAc). It receives projections from limbic structures such as the basolateral amygdala and hippocampus, and sends fibers to brain areas that interface with limbic structures with output to the extrapyramidal motor system (Alheid *et al*, 1995). A challenge is to identify the temporal processes and neural substrates that are involved in the plasticity of this neural circuit during opioid addiction. Furthermore, to develop a successful treatment for opioid addiction, the cellular events involved in the drug withdrawal episodes that enhance an opioid's incentive value to the extent that compulsive drug-taking and drug-seeking take over one's behavior need to be elucidated.

Among other neural substrates, protein kinase-mediated alterations have been reported in neural circuitries within limbic structures involved in motivational withdrawal including the NAc, amygdala (Hand *et al*, 1988; Stinus *et al*, 1990; Heinrichs *et al*, 1995), and BNST (Delfs *et al*, 2000). They have also been found within the mesencephalic and autonomic brain areas involved in somatic withdrawal including the locus coeruleus (LC) and periaqueductal gray area (Maldonado *et al*, 1992). Specifically, the $D_1$-dopaminergic receptor activity-dependent increase in cAMP-dependent protein kinase (PKA) activity that is observed following acute opioid treatment and that is enhanced in opiate withdrawal states at the NAc and dorsal striatum (Punch *et al*, 1997; Borgkvist *et al*, 2007) resulted in an increase in the phosphorylation of the transcription factor cAMP response element binding protein (pCREB) in brain regions implicated in drug addiction and in brain structures involved in behavioral plasticity associated with addiction (Shaw-Lutchman *et al*, 2002; Olson *et al*, 2005; Walters *et al*, 2005). The activation of transcription factors such as CREB can result in an increase in dynorphin expression in the NAc, leading to a reduction in drug rewards (Carlezon *et al*, 1998). This in turn counteracts the effects of ΔFosB by decreasing the transcription of dynorphin and thus contributing to drug rewards (Nestler *et al*, 2001). Although the reciprocal effects of CREB and ΔFosB on the

1 Department of Pharmacology, University of Minnesota Medical School, Minneapolis, MN, USA
2 Key Laboratory of Receptor Research, Shanghai Institute of Materia Medica and Collaborative Innovation Center for Brain Science, Chinese Academy of Science, Shanghai, China
 *Corresponding author. Tel: +1 612 624 6691; Fax: +1 612 625 8408; E-mail: kibal001@umn.edu
 **Corresponding author. Tel: +86 21 50807588; E-mail: jgliu@simm.ac.cn
 ***Corresponding author. Tel: +1 612 626 6497; E-mail: lawxx001@umn.edu
 †These authors contributed equally to this work

same gene could complicate the final addictive behavioral responses, the alteration in neuronal cAMP levels that leads to PKA-mediated changes is a direct consequence of the opioid receptor signaling process.

An opiate-induced increase in PKA-mediated phosphorylation and the activation of the cAMP/PKA/DARPP-32 signaling pathway have been shown in several brain regions. The involvement of adenylyl cyclase (AC) in chronic morphine action was demonstrated with AC5, AC1, and AC8 knockout mice, which showed the attenuation of some somatic withdrawal signs (Kim et al, 2006; Zachariou et al, 2008). An increase in cAMP levels is related to context-specific withdrawal, and several investigators have suggested that opiate withdrawal symptoms reflect classic conditioned responses (O'Brien et al, 1977; Hinson & Siegel, 1982; Schulteis et al, 2005). The inhibition of PKA activity, which disrupts memory reconsolidation, results in the weakening of motivational withdrawal (Taubenfeld et al, 2010). In fact, an increase in GluR1$^{S845}$ phosphorylation by PKA was observed in the NAc and amygdala during spontaneous withdrawal following heroin self-administration (Edwards et al, 2009). Although somatic withdrawal from opiates can be attenuated with the use of PKA-selective inhibitors (Punch et al, 1997), the use of such inhibitors might not be the ideal approach for the treatment of opioid addiction. Indeed, because of PKA interactions with multiple signaling pathways, PKA inhibitors attenuates a myriad of kinase actions that are unrelated to drug addiction (Parsons & Parsons, 2004; Babus et al, 2011; Sen & Johnson, 2011). In summary, in acute pain conditions, opioids induce analgesia by diminishing neuronal excitability by triggering intracellular signaling events that leads to the inhibition of adenylyl cyclase (AC) activity, activation of inwardly rectifying $K^+$ current, and/or inhibition of $Ca^{2+}$ conductance (Law et al, 2000). When pain persists, chronic opioid exposure not only leads to a blunting of these intracellular responses but also results in a compensatory increase in intracellular cAMP levels and AC activity in response to the excessive action of the agonist (Zhang et al, 2009, 2013). Upon the removal of the opioid from the cellular environment or the addition of an antagonist such as naloxone, the compensatory increase in AC activity becomes particularly significant and unopposed and contributes to the activation of neurons during withdrawal (Nestler, 1997). This AC superactivation phenomenon, generally referred to in the literature as the upregulation of the cAMP pathway (Nestler, 2004), has been well established as being one of the accepted molecular mechanisms of drug dependence and withdrawal (Koob & Bloom, 1988; Nestler, 2004).

Adenylyl cyclase superactivation has been demonstrated after in vivo opioid administration and subsequent naloxone injection in neurons in several regions of the rat brain, including the LC (Nestler & Aghajanian, 1997; He et al, 2009). The compensatory increase in AC activity is isozyme-specific (Avidor-Reiss et al, 1996, 1997; Yang et al, 2016). AC1, AC5, AC6, and AC8 are the isoforms that have been demonstrated to be superactivated after chronic morphine exposure in transfected cells (Avidor-Reiss et al, 1996, 1997). In mice, AC1, AC5, and AC8 are more specifically implicated in the behavioral expression of withdrawal (Kim et al, 2006; Zachariou et al, 2008).

Recently, we delineated a non-canonical signaling pathway that can account for the observed superactivation of AC in cell model systems (Zhang et al, 2013). During acute morphine treatment, the μ-opioid receptor (MOR) interacts with $G_{\alpha i2}$ (Zhang et al, 2006) and

causes an inhibition of AC activity and a decrease in the intracellular cAMP level. Under prolonged treatment, the MOR non-canonical signaling pathway is activated. It involves the recruitment of Src kinase by the MOR signal complex within which Src kinase is activated. The activated Src phosphorylates MOR at Tyr$^{336}$, although it remains to be demonstrated whether the phosphorylation is direct or indirect. In turn, the phosphorylated Tyr$^{336}$ serves as a docking site for growth factor receptor-bound protein (Grb)/son of sevenless (SOS), leading to the recruitment and activation of Ras. Activated Ras binds to Raf-1 and recruits Src kinase that will activate Raf-1 in the membrane proximity resulting in Raf-1 increase in activity (Baccarini, 2005). This leads to the subsequent phosphorylation and activation of AC5/6 by Raf-1 or, in other words, the AC superactivation. Hence, the direct or indirect phosphorylation of MOR at Tyr$^{336}$ by Src kinase serves as the switch for converting a classic $G_i/G_o$-coupled receptor into a receptor tyrosine kinase-like entity, resulting in a non-canonical signaling pathway even after the canonical $G_i/G_o$ signaling has been attenuated. This non-canonical signaling pathway seems to be responsible for the switching from MOR-mediated initial AC inhibition to subsequent AC activation (Zhang et al, 2013).

If the PKA-mediated neural plasticity within the addiction circuit that is a direct result of AC superactivation is a probable cause for compulsive drug use due to withdrawal/negative reinforcement, then PKA-mediated neural plasticity can be modulated by regulating the pre-required Src-mediated phosphorylation of MOR at Tyr$^{336}$. Whether such changes in Src activity and MOR phosphorylation occur in vivo during opiate withdrawal is unknown. To establish the relationship between Tyr$^{336}$ phosphorylation and opiate withdrawal, we developed an antigen-purified rabbit polyclonal antibody against the phosphorylated Y336 residue of MOR. Here, we demonstrate that MOR is phosphorylated at Tyr$^{336}$ during naloxone-precipitated withdrawal. Using Fyn$^{-/-}$ mice and the Src kinase inhibitor AZD0530, we show that the phosphorylation of MOR is mediated by the Src family kinases (SFKs). Furthermore, we reveal that the naloxone-precipitated somatic withdrawal signs observed after chronic morphine treatment are dependent on the Src-mediated phosphorylation of MOR at Tyr$^{336}$. Our findings suggest that small molecules that modulate the Src-mediated phosphorylation of MOR might prevent the transition to the withdrawal/negative reinforcement stage of the opiate addiction cycle.

## Results

### MOR is phosphorylated at Tyr$^{336}$ during naloxone-precipitated withdrawal

To demonstrate that MOR is phosphorylated at Tyr$^{336}$, we developed an affinity-purified polyclonal antibody against phosphorylated Tyr$^{336}$ (anti-pMOR$^{Y336}$). The specificity of the antibody was demonstrated in clonal cell lines expressing a wild-type (WT) or Y336F mutant MOR. As shown in Appendix Figs S1A–D and S2A–D, this antibody detected the phosphorylated, but not the non-phosphorylated, Y336 residue of MOR in both Western and immunofluorescence analyses. Although the peptide sequence used to immunize the rabbits (NPVLYAFLDEN) is conserved in all three of the opioid receptors, anti-pMOR$^{Y336}$ did not detect the non-phosphorylated

MOR or KOR, and exhibited only slight reactivity toward the phosphorylated DOR. The antibody is specific because it does not detect any pY336 in HEK293 cells expressing the Y336F mutant in both Western and immunofluorescence analyses (Appendix Figs S1A–D and S2A–D). Anti-pMOR$^{Y336}$ is specific also because of the disappearance of pY336 immunoreactivity when the pMOR$^{Y336}$ antibody was pre-incubated with the immunoprecipitated MOR complex that had been extracted from the LC of morphine-dependent WT mice undergoing naloxone-precipitated withdrawal (Fig 1B) and with the corresponding phospho-peptide (Fig 2A). The pSrc$^{Y416}$ antibody is well characterized and used in many reported studies, especially for Western blot (Koreckij *et al*, 2009; Megison *et al*, 2014).

In order to establish the causal relationship between the increase in pMOR$^{Y336}$ and naloxone-precipitated somatic opiate withdrawal, we examined whether anti-pMOR$^{Y336}$ can detect an increase in MOR phosphorylation in the LC. The LC has been reported to be the primary anatomical site responsible for the expression of the motor components of opiate withdrawal, such as jumping, rearing, and locomotor activity, in studies using electrolytic lesions, PKA inhibitors, or PKA activators (Maldonado *et al*, 1992; Maldonado & Koob, 1993; Punch *et al*, 1997). As shown in Fig 1A, increases in both the pSrc$^{Y416}$ and the pMOR$^{Y336}$ immunofluorescence signals in the LC were observed in mice implanted with a morphine pellet only after naloxone-precipitated withdrawal. The increases were not observed in mice with morphine pellets in place or in mice implanted with placebo pellets and subjected to naloxone treatment. Similar increases in both pSrc$^{Y416}$ and pMOR$^{Y336}$ levels were observed in mice chronically treated with morphine according to progressive dose injection methods, although a lower level of pMOR$^{Y336}$ immunofluorescence was observed (data not shown). A colocalization analysis indicated that $79.4 \pm 12.0\%$ of the tyrosine hydroxylase-positive (TH$^+$) cells exhibited both pSrc$^{Y416}$ and pMOR$^{Y336}$ immunofluorescence, suggesting that the increase in Src activity and MOR$^{Y336}$ phosphorylation occurred in TH$^+$ neurons within the LC during somatic withdrawal ($F_{(2,19)} = 303.9$, $P < 0.0001$, one-way ANOVA; Fig 1C). Other regions such as the hippocampus also exhibited similar increases in pSrc$^{Y416}$ and pMOR$^{Y336}$ levels during naloxone-precipitated somatic withdrawal (Fig EV1C). The colocalization of the pSrc$^{Y416}$ and pMOR$^{Y336}$ immunofluorescent signals was observed at proximity of the cell surface at the CA1 and CA3 regions of the hippocampus, suggesting that Src may be recruited to the vicinity of the surface-localized MOR (Fig EV1D).

The increase in the pMOR$^{Y336}$ immunofluorescent signal requires the presence of MOR and Src. MOR$^{-/-}$ mice implanted with a morphine pellet and subjected to naloxone-precipitated opiate withdrawal showed a minimal increase in pSrc$^{Y416}$ and pMOR$^{Y336}$ levels (Fig EV1A). The residual $3.17 \pm 0.53\%$ colocalization of pSrc$^{Y416}$ and pMOR$^{Y336}$ in MOR$^{-/-}$ mice (Fig 1C) might represent the cross-reactivity of the anti-pMOR$^{Y336}$ toward pDOR that was detected in the Western and immunofluorescence analyses (Appendix Figs S1A–D and S2A–D). As for the requirement of Src, five members of the SFK, N-Src, Fyn, Lyn, Yes, and Lck, are present at substantial levels in the adult mammalian brain especially in the amygdala, NAc, VTA, and midbrain (Ohnishi *et al*, 2011). However, Fyn is the major subtype that has been shown to be implicated in multiple neural functions, including the regulation of NMDA receptor activity (Miyakawa *et al*, 1997) and behavior in response to ethanol exposure (Boehm *et al*, 2003). Thus, Fyn$^{-/-}$ mice were used to examine

whether naloxone-precipitated somatic withdrawal after chronic morphine pellet implantation would elicit a similar increase in pSrc$^{Y416}$ and pMOR$^{Y336}$ levels. As summarized in Fig 1A, an increase in pSrc$^{Y416}$ immunofluorescence was observed in the LC of the mice implanted with the morphine pellets and subjected to naloxone treatment compared to that observed in the LC of mice implanted with the placebo pellets and similarly treated with naloxone. Such an increase was anticipated since 50% of the total Src kinase enzymatic activity remained in the Fyn$^{-/-}$ mice brain homogenates. Nevertheless, as with the increase in pSrc$^{Y416}$, a minimal increase in pMOR$^{Y336}$ immunofluorescence was observed, and only $13.9 \pm 3.2\%$ of this signal colocalized with pSrc$^{Y416}$ in Fyn$^{-/-}$ mice subjected to naloxone-precipitated opiate withdrawal (Fig 1C). It is likely that Fyn is the major Src subtype that contributes to the increase in pMOR$^{Y336}$ levels during somatic withdrawal.

## Src is recruited to the MOR signaling complex during naloxone-precipitated withdrawal

For the Src-mediated phosphorylation of MOR to occur, the tyrosine kinase has to be recruited in the vicinity of the receptor (Zhang *et al*, 2013). We conjugated a peptide affinity-purified MOR rabbit polyclonal antibody directed against the MOR N-terminal sequence SHVDGNQSDPCGLNRTGLGGNDSLCPQTGSPSMVTHRD to Dynabeads$^\circledR$M-270 Epoxy beads (Life Technologies) to immunoprecipitate (IP) the receptor complex. The specificity of the MOR antibody was demonstrated by its inability to IP any receptor complex in MOR$^{-/-}$ mice. AC superactivation has been shown to be independent from agonist-induced receptor internalization and to require the location of both MOR and the G$_{\alpha i2}$ proteins at lipid rafts (Zhao *et al*, 2006). Since immunoprecipitated MOR was reported to be phosphorylated at Tyr$^{336}$ by Src kinase within lipid rafts (Zhang *et al*, 2009), the pSrc$^{Y416}$ or pMOR$^{Y336}$ detected with our Western blots are assumed to be from MOR-G$_{\alpha i2}$-Src signaling complex located within lipid rafts and pulled down with the MOR N-terminus antibody. As shown in Fig 2A, the IP from the midbrain extracts from mice implanted with a placebo pellet revealed the presence of basal pSrc$^{Y416}$ but not of pMOR$^{Y336}$ (lane 1). The IP from the extracts from mice implanted with a 75-mg morphine pellet for 4 days and subjected to naloxone-precipitated withdrawal showed an increase in the amount of pSrc$^{Y416}$ associated with MOR and phosphorylated MOR$^{Y336}$ (Fig 2A, lane 3). No detectable increase in either pSrc$^{Y416}$ or pMOR$^{Y336}$ was observed in extracts from mice that were implanted with a morphine pellet but did not undergo naloxone-precipitated withdrawal (Fig 2A, lane 5). The increases in pMOR$^{Y336}$ and pSrc$^{Y416}$ levels within the receptor complex after naloxone-precipitated withdrawal were determined to be $5.69 \pm 1.38$-fold and $3.86 \pm 1.10$-fold, respectively, over the placebo control (Fig 2B and C; $n = 3$ blots, $P = 0.001$ in both cases). Hence, the MOR non-canonical signaling pathway exists in the mouse brain and parallels that observed in cell models.

## Inhibition of Src kinase activity with AZD0530 blocked signs of naloxone-precipitated somatic withdrawal

If Src-mediated increase in pMOR$^{Y336}$ levels is key to induction of naloxone-precipitated withdrawal, then the *in vivo* inhibition of Src activity should blunt withdrawal. Narita *et al* (2006) reported that

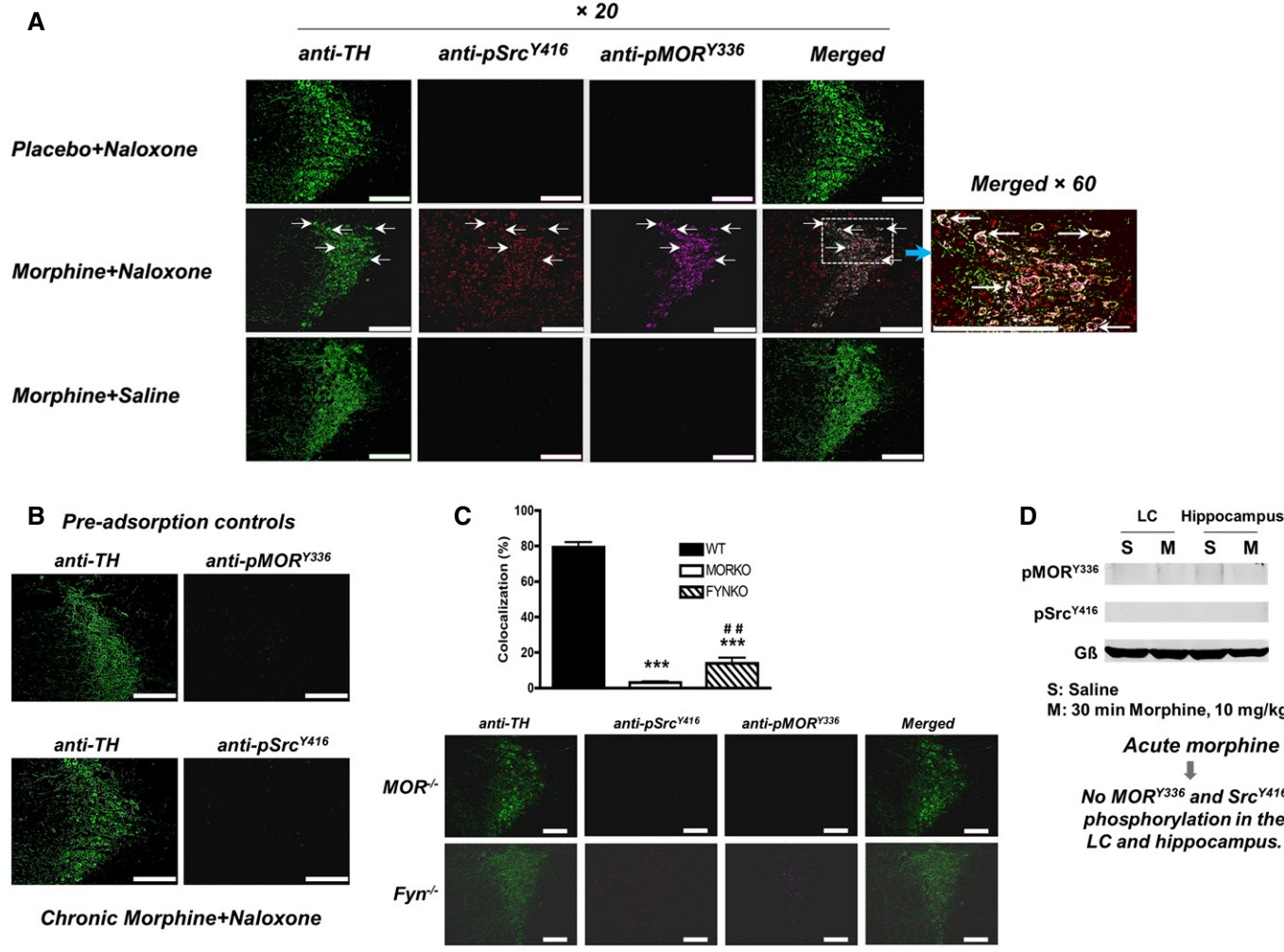

**Figure 1. Increase in Src activation and MOR phosphorylation at Tyr^336 during naloxone-precipitated withdrawal.**

WT mice were implanted with either placebo or morphine pellets as detailed in the Materials and Methods.

A   Low (×20) and high-magnification (×60) photomicrographs showing the colocalization of pMOR^Y336 and pSrc^Y416 in the TH^+ neurons of the LC (arrows) from a placebo-and-naloxone-treated mouse (Placebo + Naloxone), a morphine-and-naloxone-treated mouse (Morphine + Naloxone), and a morphine-and-saline-treated (Morphine + Saline) mouse. The green, red, and magenta colors represent TH, pSrc^Y416, and pMOR^Y336, respectively. Scale bar = 70 μm.

B   The specificity of the pMOR^Y336 and pSrc^Y416 antibodies was confirmed by the disappearance of pMOR^Y336 and pSrc^Y416 immunoreactivity when pMOR^Y336 and pSrc^Y416 antibodies were pre-incubated with the immunoprecipitated MOR complex that had been extracted from the LC of morphine-dependent WT mice with naloxone-precipitated withdrawal (Morphine + Naloxone). Scale bar = 70 μm.

C   The bar graph shows the colocalization analysis of the WT, MOR^−/−, and Fyn^−/− mice (Fig EV1). The percentage of colocalized pMOR^Y336, pSrc^Y416, and TH results from the quantification in three mice/genotype, four slices/mouse. Significant differences among the groups (WT, MOR^−/−, and Fyn^−/−) were determined using one-way ANOVA, followed by Duncan's *post hoc* comparison. ***$P$ = 0.0001 MOR^−/− versus WT; ***$P$ = 0.0001 Fyn^−/− versus WT; ^##$P$ = 0.002729 significant differences between MOR^−/− and Fyn^−/−. The photomicrographs show minimal colocalization between pMOR^Y336 and pSrc^Y416 in the TH^+ neurons of the LC from MOR^−/− or Fyn^−/− mice treated with chronic morphine (pellets for 3 days) and injected with naloxone after morphine pellets removal on the 4^th day. The green, red, and magenta colors represent TH, pSrc^Y416 and pMOR^Y336, respectively. Data are presented as means ± SEM. Scale bar = 70 μm.

D   The phosphorylation of MOR^Y336 (pMOR^Y336) and Src^Y416 (pSrc^Y416) was not detected within the MOR complex in the LC and the hippocampus of WT mice after acute injection of morphine (30 min, 10 mg/kg). An affinity-purified polyclonal antibody against the MOR N-terminus (conjugated to Dynabeads) was employed to IP pSrc and pMOR as described in the Materials and Methods and Fig 2A. The immunoblots were probed with anti-pSrc^Y416 (1:1,000), anti-pMOR^Y336 (1:500), or anti-G_β (Santa Cruz sc-378, 1:500).

Source data are available online for this figure.

the intracerebroventricular (i.c.v.) injection of the water-insoluble Src kinase inhibitor PP2 can block the reward effect and hyperloco-motion induced by morphine. Because the stereotaxic injection of the inhibitor PP2, which is dissolvable only in organic solvents such as DMSO, damaged the LC, we used the water-soluble Src kinase inhibitor Saracatinib (AZD0530). Moreover, AZD0530 is one of the four SFK inhibitors [including Dasatinib, Bosutinib (SKI-606), and KX2–391] that are currently undergoing clinical evaluation in oncology (Puls *et al*, 2011), and has >250-fold selectivity for the SFK over other tyrosine kinase families (Green *et al*, 2005, 2009). AZD0530

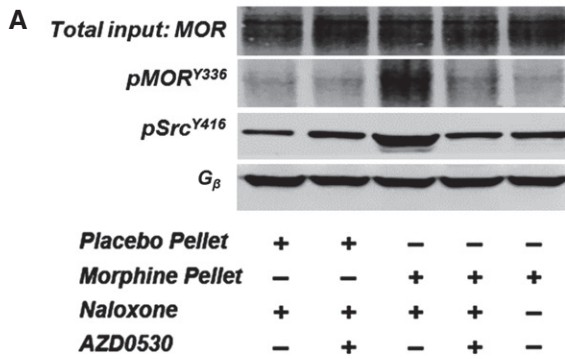

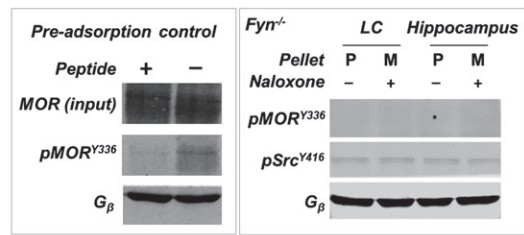

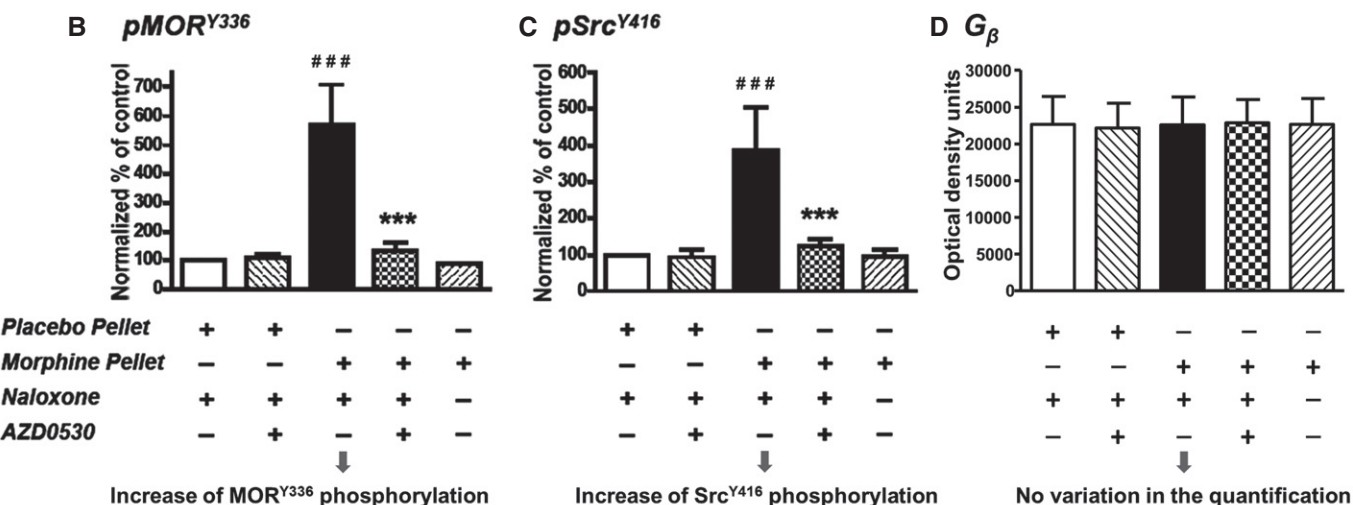

**Increase of MOR$^{Y336}$ phosphorylation during naloxone-precipitated withdrawal.**

**Increase of Src$^{Y416}$ phosphorylation during naloxone-precipitated withdrawal.**

**No variation in the quantification of the internal control G$_\beta$.**

**Figure 2. Src is recruited to and activated within the MOR complex during naloxone-precipitated withdrawal.**

The midbrain was dissected from mice that had been implanted with placebo or morphine pellets and i.c.v. injected with saline or AZD0530.

A An affinity-purified polyclonal antibody against the MOR N-terminus (conjugated to Dynabeads) was employed to IP pSrc and pMOR as described in the Materials and Methods. The immunoblots were probed with anti-pSrc$^{Y416}$ (1:1,000), anti-pMOR$^{Y336}$ (1:500), or anti-G$_\beta$ (Santa Cruz sc-378, 1:500). In the middle panel, a Western blot analysis shows the disappearance of the pMOR$^{Y336}$ immunoreactivity when the pMOR$^{Y336}$ antibody was pre-incubated with the corresponding phospho-peptide (GeneTex PJ90076 NPVL(pY)AFLDENC). This pre-adsorption control was performed with the immunoprecipitated MOR complex that had been extracted from the LC of morphine-dependent WT mice with naloxone-precipitated withdrawal. On the right, another Western blot shows residual Src activities in the Fyn$^{-/-}$ mice (Src needs to be phosphorylated at Tyr$^{416}$ before activation). The reduced bands correlate with the sparsely detectable pSrc$^{Y416}$ immunoreactivity in the photomicrographs of Figs 1C and EV1B. This should be a clear indication of the specificity of the anti-pSrc$^{Y416}$.

B The bar graphs show the densitometric quantification and statistical analysis of the changes in pMOR$^{Y336}$ from the experiment presented in (A, left panel).

C The bar graphs show the densitometric quantification and statistical analysis of the changes in pSrc$^{Y416}$ from the experiment presented in (A, left panel).

D The bar graphs show no significant differences in the densitometric quantification of G$_\beta$ (from the experiment presented in A, left panel) among the conditions (the internal control, $n = 3$ mice).

Data information: Data are presented as means ± SEM. In (B and C), the density of each band was normalized against G$_\beta$ (the internal control), and the result from the placebo with naloxone group (lane 1) was set to 100%. The experiments were repeated three times. Significant differences between the various treatment groups ($n = 3$ mice/group) (lanes 2–5) and the control ($n = 3$ mice) (lane 1) were determined using Student's $t$-test. $^{###}P = 0.001$ Placebo + Naloxone versus MS + Naloxone in both (B and C), $^{***}P = 0.00086$ and $^{***}P = 0.00077$ MS + Naloxone versus MS + Naloxone + AZD in (B and C), respectively.

inhibits Src with a potency of 2 nM and inhibits Abl at 30 nM (Hennequin *et al*, 2006). However, the selective inhibition of Src over Abl when mice are i.c.v. injected with 50 µg of AZD0530 has been demonstrated. The observed increase in Src enzymatic activity that occurs during naloxone-precipitated withdrawal was blunted after the i.c.v. injection of AZD0530, while the observed increase in Abl activity was not altered (Fig 3). A parallel decrease in pSrc$^{Y416}$ associated with the MOR complex and the inhibition of pMOR$^{Y336}$

was also observed in mice i.c.v. injected with 50 µg of AZD0530 (Fig 2A, lane 4). It is likely that *in vivo* i.c.v. doses of AZD0530 < 50 µg inhibit Src kinase activity without affecting Abl. The observed alteration in pMOR$^{Y336}$ levels is the direct consequence of Src kinase activity modulated by AZD0530.

The ability of AZD0530 to selectively inhibit Src kinase activity allowed us to investigate whether the Src-mediated phosphorylation of MOR$^{Y336}$ is required for naloxone-precipitated withdrawal. Mice

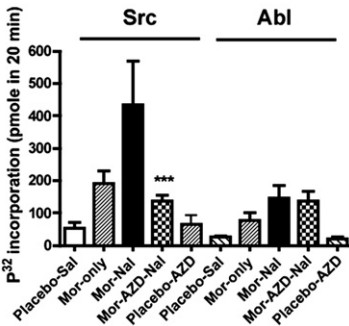

**Figure 3.  Selective inhibition of Src but not Abl activity in mice i.c.v. injected with AZD0530.**

Mice were implanted either with placebo or morphine, and AZD0530 or saline was then injected i.c.v. Then, the mice were s.c. injected with or without naloxone (Nal) as detailed in the Materials and Methods. The midbrains of the mice in each treatment group were dissected and used in a Src or Abl kinase activity assay as described (Zhang *et al*, 2013). ***$P$ = 0.00093 for Mor-AZD-Nal treatment compared with placebo-Sal as determined by Students *t*-test. Each experiment was repeated three times. Data are presented as means ± SEM.

Source data are available online for this figure.

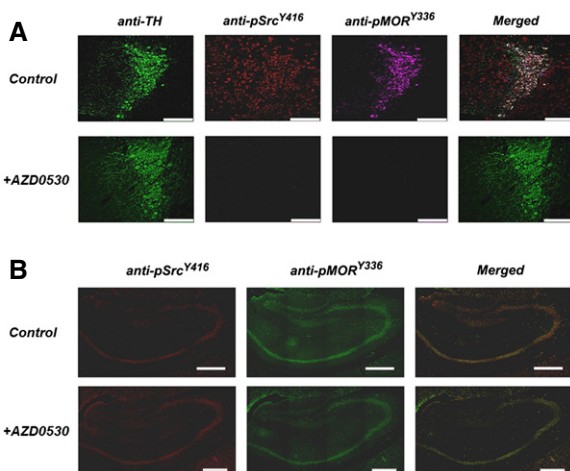

**Figure 4.  Inhibition of Src activity and MOR$^{Y336}$ phosphorylation in the LC, but not in the hippocampus, of mice stereotaxically injected in the LC with AZD0530.**

WT mice were either implanted with placebo or morphine pellets for 4 days. Saline or AZD0530 (50 μg/side) was injected into the LC 30 min prior to naloxone administration as detailed in the Materials and Methods.

A  Immunostaining of TH (green), pSrc$^{Y416}$ (red), or MOR$^{Y336}$ (magenta) in the LC following the injection of saline (upper) or AZD0530 (lower) into the LC. Scale bar = 70 μm.

B  Immunostaining of pSrc$^{Y416}$ (red) or MOR$^{Y336}$ (green) in the hippocampus following the injection of saline (upper) or AZD0530 (lower) into the LC. The yellow denotes overlap. Scale bar = 100 μm.

Source data are available online for this figure.

that had been implanted with a morphine pellet for 4 days were subjected to naloxone-precipitated withdrawal. Thirty minutes prior to the injection of naloxone, 50 μg of AZD0530 was injected into the third ventricle via an implanted cannula. The injection of AZD0530 blocked the increase in the immunofluorescence of pSrc$^{Y416}$ and pMOR$^{Y336}$ in the LC after naloxone-precipitated opiate withdrawal (Fig 4A). Several somatic withdrawal signs such as diarrhea, jumping, mastication, paw tremors, and wet dog shakes (WDS) were significantly blunted after the AZD0530 injection; others such as grooming and body weight change were not altered by AZD0530 injection (Figs 5A and EV2). The number of jumps observed after naloxone injection decreased from 89.7 ± 27.0 in control mice to 23.7 ± 13.0 in mice injected with AZD0530 ($F_{(2,21)}$ = 9.29, $P$ = 0.0013, one-way ANOVA; $P$ = 0.006141, Duncan's *post hoc* test). A significant decrease in global withdrawal scores from 101 ± 12.8 to 21 ± 6.7 ($F_{(2,21)}$ = 45.01, $P$ < 0.001, one-way ANOVA; $P$ = 0.000147, Duncan's *post hoc* test) was observed in the presence of AZD0530 (Fig 5A). Similarly, the AZD0530-mediated inhibition of naloxone-precipitated withdrawal signs was observed in mice chronically treated with progressive morphine doses. The effect of AZD0530 was dose-dependent and blunted the signs of withdrawal when stereotaxically injected into the LC (Appendix Fig S3). The stereotaxic injection of 5 μg of AZD0530 bilaterally into the LC eliminated the pSrc$^{Y416}$ and pMOR$^{Y336}$ immunofluorescent signals in the LC after naloxone-precipitated withdrawal (data not shown), but it did not affect the signals in other brain areas such as the hippocampus (Fig 4B). The stereotaxic injection of AZD0530 into the LC also blunted several, but not all, the naloxone-precipitated withdrawal signs. The affected withdrawal signs included: jumping, mastication, paw tremors, and WDS (Figs 5B and EV3). A significant decrease in the global withdrawal score from 60 ± 4.3 ($n$ = 11) to 27 ± 6.9 ($n$ = 13) (unpaired Student's *t*-test, $P$ = 0.000656) was observed in the presence of AZD0530 (Fig 5B). The somatic withdrawal signs inhibited by the stereotaxic injection of AZD0530 are in accordance with the withdrawal signs shown to be associated with the LC (Maldonado *et al*, 1992). Thus, the

AZD0530-mediated selective inhibition of global Src kinase activity blocked increases in pSrc$^{Y416}$ and pMOR$^{Y336}$ levels and subsequently diminished some of the naloxone-precipitated somatic withdrawal signs.

## Naloxone-precipitated somatic withdrawal signs are dependent on Fyn levels

The central role of Src activity and naloxone-precipitated somatic withdrawal was further demonstrated in Fyn$^{-/-}$ mice. WT, heterozygous, and homozygous Fyn$^{-/-}$ mice were implanted with morphine pellets for 4 days and subjected to naloxone-precipitated withdrawal. As summarized in Figs 1C and EV1B, increases in pSrc$^{Y416}$ and pMOR$^{Y336}$ levels after naloxone-precipitated withdrawal were not observed in Fyn$^{-/-}$ mice. A parallel decrease in naloxone-precipitated withdrawal signs was observed. Some somatic withdrawal signs such as diarrhea, jumping, mastication, paw tremors, and WDS were blunted in the Fyn$^{-/-}$ mice (Figs 5C and EV4). These alterations in somatic withdrawal signs in the Fyn$^{-/-}$ mice are similar to those observed with AC1$^{-/-}$ but not AC8$^{-/-}$ mice (Zachariou *et al*, 2008). The incomplete blockade of somatic withdrawal signs could be due to the presence of other Src isoforms, as suggested by the presence of residual Src kinase activity and pSrc$^{Y416}$ immunofluorescence within the LC structure of the Fyn$^{-/-}$ mice (Fig EV1B). Nonetheless, the same signs that were blunted in the WT mice injected with AZD0530 were also reduced in the Fyn$^{-/-}$ mice. The blunting of the withdrawal signs appears to be dependent on gene dosage. Overall global withdrawal signs were

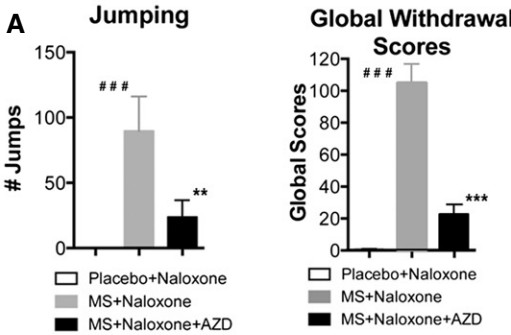

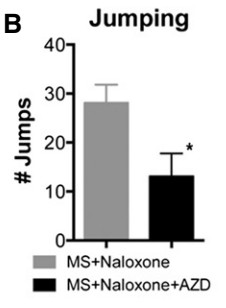 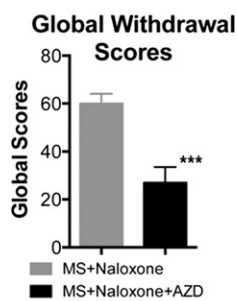

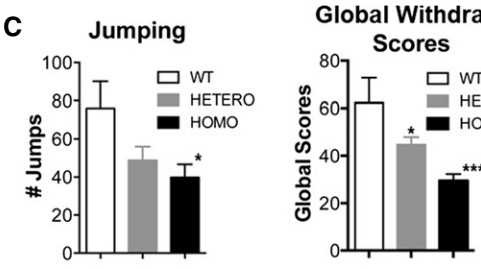

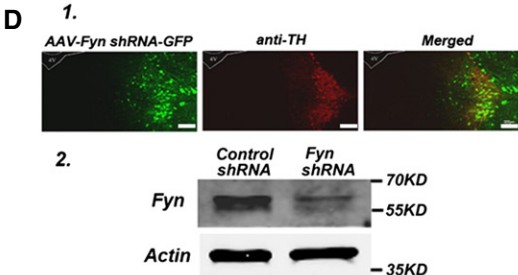

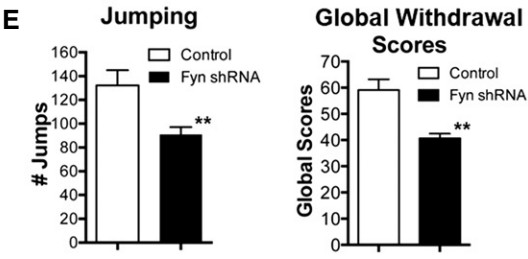

**Figure 5. The effect of Src activity or levels on naloxone-precipitated somatic withdrawal signs.**

Src activity was inhibited by either the i.c.v. or stereotaxic injection of AZD0530 into the LC. The Src level was reduced in Fyn$^{-/-}$ mice or in mice in which either Fyn shRNA lentivirus or AAV had been injected into the LC.

A The effect of AZD0530-mediated inhibition of Src activity on naloxone-precipitated somatic withdrawal behavior. Fifty micrograms of AZD0530 was i.c.v. injected into WT mice as described in the Materials and Methods. The inhibition of Src activity induces a significant decrease in naloxone-precipitated jumping, other withdrawal signs (Fig EV2), and global withdrawal scores. One-way ANOVA followed by Duncan's *post hoc* comparison revealed significant differences between the morphine-and-naloxone-treated mice (MS + Naloxone, $n = 8$) and morphine-naloxone-and-AZD0530-treated mice (MS + Naloxone + AZD, $n = 8$) in the number of jumps **$P = 0.006141$ and global withdrawal scores ***$P = 0.000147$. Significant differences between the placebo-and-naloxone-treated mice (Placebo + Naloxone, $n = 8$) and morphine-and-naloxone-treated mice (MS + Naloxone) were shown in the number of jumps $^{###}P = 0.000677$ and global withdrawal scores $^{###}P = 0.00007$.

B Effect of Src inhibition on naloxone-precipitated somatic withdrawal signs following the stereotaxic injection of AZD0530 (7.5 μg/μl) into the LC. The LC-specific inhibition of Src activity causes a significant decrease in naloxone-precipitated jumping, other withdrawal signs (Fig EV3), and global withdrawal scores. Significant differences between the morphine-and-naloxone-treated mice (MS + Naloxone, $n = 11$) and morphine-naloxone-and-AZD-treated mice (MS + Naloxone + AZD, $n = 13$) were determined using Student's *t*-test: *$P = 0.0190$ for jumping and ***$P = 0.000656$ for global withdrawal scores.

C The reduction in naloxone-precipitated withdrawal signs in heterozygous (HETERO) and homozygous (HOMO) Fyn$^{-/-}$ mice ($n = 6$/group). The jumping and global withdrawal scores were significantly blunted in the HOMO Fyn$^{-/-}$ mice. One-way ANOVA followed by Duncan's *post hoc* test revealed significant differences between the WT and HOMO mice in the number of jumps *$P = 0.017939$ and the global withdrawal scores ***$P = 0.000409$. There are also significant differences between the WT and HETERO mice in the global withdrawal scores *$P = 0.022127$.

D The reduction in Fyn levels following the stereotaxic injection of Fyn shRNA virus. (1.) Immunofluorescence analysis showing the colocalization of AAV-Fyn-shRNA-GFP (green) with TH (red), a marker for adrenergic neurons in the LC. Scale bar = 200 μm. (2.) Representative Western blot image of the *in vivo* knockdown of Fyn expression.

E The knockdown of Fyn in the LC significantly attenuated naloxone-precipitated morphine withdrawal signs. One microliter of Fyn shRNA AAV virus (titer: $2.09 × 10^{12}$ v.g./ml) was injected bilaterally into the LC ($n = 12–13$) as described in the Materials and Methods. Fyn knockdown induced a significant decrease in naloxone-precipitated jumping, other withdrawal signs (Fig EV5), and global withdrawal scores. Student's *t*-tests showed significant differences between the control (GFP-transferred) and Fyn shRNA-transferred WT mice in the number of jumps **$P = 0.009429$ and global withdrawal scores **$P = 0.001723$.

Data information: The values in graphs are expressed as the means ± SEM.

significantly decreased from $62.3 ± 10.5$ in the WT mice to $29.5 ± 2.8$ in the homozygotes ($F_{(2,15)} = 11.40$, $P = 0.000978$, one-way ANOVA; $P = 0.000409$, *post hoc*; Fig 5C).

The probable involvement of Fyn in naloxone-precipitated withdrawal can be further substantiated by the decrease in total Fyn levels following injection of Fyn shRNA. Regardless of the viral vehicle, either $5.0 × 10^8$ TU/ml for lentivirus or $2.1 × 10^{12}$ TU/ml for AAV2/9, the injection of the Fyn shRNA virus into the LC (Fig 5D, panel 1.) resulted in a $70.8 ± 4.5\%$ ($n = 8$, $P = 0.01$) decrease in total Fyn levels as demonstrated by Western blot analysis (Fig 5D, panel 2.). When naloxone-precipitated withdrawal signs were monitored in mice injected with Fyn shRNA virus and compared to those injected with GFP virus, significant reductions in several withdrawal

signs such as jumping, mastication, and paw tremors were observed (Figs 5E and EV5). The reduction in jumping was significant, but was much less than that observed following AZD0530 injection or in the Fyn$^{-/-}$ mice (GFPv = $132.2 \pm 12.77$, $n = 13$; Fyn shRNAv = $91.0 \pm 6.12$, $n = 12$; $P = 0.009429$ unpaired Student's $t$-test). Furthermore, the WDS somatic withdrawal sign that was significantly reduced in AZD0530-injected mice or in Fyn$^{-/-}$ homozygotes remained unchanged in the Fyn shRNA-injected mice. This discrepancy could be due to incomplete knockdown resulting in a residual level of Fyn in the LC (70.8% decrease). Nevertheless, there was a significant decrease in the global withdrawal scores, from $59.1 \pm 4.1$ to $40.7 \pm 1.8$ (GFPv, $n = 13$; Fyn shRNA, $n = 12$; $P = 0.001723$ unpaired Student's $t$-test).

### Phosphorylation of MOR at Tyr$^{336}$ is a prerequisite for naloxone-precipitated withdrawal

The role of the Src-mediated phosphorylation of MOR at Tyr$^{336}$ in naloxone-precipitated withdrawal was further substantiated in MOR$^{-/-}$ mice. The absence of any morphine dependence or somatic withdrawal signs in the MOR$^{-/-}$ mice is unequivocal. If the MOR non-canonical signaling pathway is critical during naloxone-precipitated withdrawal, we should be able to delineate differences in the somatic withdrawal in MOR$^{-/-}$ mice after the WT or the Y336F mutant MOR is reintroduced into the brain regions involved in opiate withdrawal. Lentiviruses in which a TH promoter fragment (kindly provided by Uwe Maskos, Pasteur Institute, Paris, FR) was used to control the expression of the transgene in the TH$^{+}$ neurons of the LC were constructed (Toiu *et al*, 2010). Equal amounts of lentivirus, $3.82 \times 10^5$ TU/ml with the TH promoter driving the expression of either GFP (TH-GFPv), WT MORGFP (TH-MORGFPv), or mutant MORY336FGFP (TH-Y336FGFPv), were injected bilaterally into the LC 1 week prior to the implantation of a morphine pellet. The mice were subsequently subjected to naloxone-precipitated withdrawal 4 days later. Immunofluorescence analysis indicated that the expression of the transgenes was localized within the LC structure (Appendix Fig S4A–F). $46.34 \pm 15.36\%$ and $47.82 \pm 13.68\%$ of the TH$^{+}$ neurons expressed the TH-MORGFPv and mutant TH-Y336FGFPv, respectively. All the somatic withdrawal signs we graded and monitored, we observed the restoration of withdrawal jumping and WDS in all the MOR$^{-/-}$ mice that had been stereotaxically injected with the TH-MORGFPv (Fig 6). Although the naloxone-precipitated jumping in the MOR$^{-/-}$ mice injected with the TH-MORGFPv was significantly lower than that observed in the WT mice treated similarly ($79.67 \pm 17.61$ jumps in WT mice ($n = 6$) vs. $13.25 \pm 5.46$ jumps in MOR$^{-/-}$ mice with TH-MORGFPv ($n = 8$) ($P = 0.001564$, Duncan's *post hoc* test; $F_{(3,22)} = 17.35$, $P = 0.000005$, one-way ANOVA with "MOR gene expression" as the variable), withdrawal jumping was not observed in MOR$^{-/-}$ mice that had been stereotaxically injected with either TH-GFPv or TH-Y336FGFPv (Fig 6). The restoration of the WDS in MOR$^{-/-}$ mice appeared to be more robust than the restoration of the jumping. The MOR$^{-/-}$ mice that had been stereotaxically injected with the WT MORGFP virus exhibited a greater number of shakes ($20.25 \pm 4.59$ shakes) than the WT mice treated with MS ($15 \pm 2.99$ shakes) (Fig 6; $P = 0.393488$, Duncan's *post hoc* test). The MOR$^{-/-}$ mice injected with TH-GFPv did not exhibit any restoration of WDS. However, WDS were observed in five out of six

MOR$^{-/-}$ mice that had been injected with TH-Y336FGFPv (and in no mice that had been injected with TH-GFPv). The average was $5.00 \pm 1.77$ shakes in TH-Y336FGFPv mice, which is significantly lower than that observed in WT ($P = 0.016420$, Duncan's *post hoc* test) or MOR$^{-/-}$ mice injected with TH-MORGFPv ($P = 0.005544$, Duncan's *post hoc* test; Fig 6; $F_{(3,22)} = 8.19$, $P = 0.000761$, one-way ANOVA with "MOR gene expression" as the variable). MOR gene expression levels were similar between the TH-MORGFPv- and TH-Y336FGFPv-transferred MOR$^{-/-}$ mice as verified by RT–qPCR (Fig 7A). Regression analyses of the MOR mRNA levels correlating with the withdrawal scores were performed with several MOR-transferred MOR$^{-/-}$ mice that were taken randomly from the experiment presented in Fig 6. However, the sample is not sufficiently large to detect precisely the relationship between the WT or mutant MOR mRNA levels and the corresponding scores of each withdrawal signs (Fig 7B and C), although there seems to be a strong correlation between the level of MOR gene expression and the number of WDS in the TH-MORGFPv-transferred MOR$^{-/-}$ mice. Hence, the mutation of Tyr$^{336}$ to Phe, which blocked Src-mediated phosphorylation and the initiation of the MOR non-canonical signaling pathway, attenuated naloxone-precipitated withdrawal.

## Discussion

The increase in AC activity above the basal level during naloxone-precipitated withdrawal has long been a working hypothesis for the PKA activation that leads to alterations in the DARPP-32 signaling pathway in various brain regions and to the subsequent behavioral responses observed during opiate withdrawal. The exact mechanism of AC superactivation was not established until our recent observations that identified the requirement for the Src-mediated phosphorylation of MOR at Tyr$^{336}$. The phosphorylation of MOR resulted in the formation of a signaling complex that consisted of Grb/SOS/Ras/Raf-1 and resembled that of the tyrosine kinase receptor (Zhang *et al*, 2013). Such a non-canonical MOR signaling pathway was demonstrated in a cell model system where receptor mutagenesis studies were performed in conjunction with the use of a p-Tyr pan-antibody. In the current studies, we could demonstrate the *in vivo* phosphorylation of MOR at Tyr$^{336}$ with the affinity-purified antibody anti-pMOR$^{Y336}$. The specificity of this antibody was clearly demonstrated by its inability to detect any MOR phosphorylation in HEK293 cells expressing a MORY336F mutant and by the pre-adsorption controls (Figs 1B and 2A). Although this antibody could detect the phosphorylation of DOR at Tyr$^{318}$ in HEK293 cells due to the peptide sequence NPVLpY that was used to develop the antibody, the minimal signal detected in MOR$^{-/-}$ mice after chronic morphine treatment (Fig EV1A) confirms our conclusion that anti-pMOR$^{Y336}$ is indeed detecting the phosphorylation of MOR at Tyr$^{336}$. A definitive conclusion regarding the phosphorylation of MOR at Tyr$^{336}$ could come from the mass spectrometric analysis of the phospho-peptides isolated from the enzymatic digest of purified MOR. However, due to the location of Tyr$^{336}$, obtaining an optimal digest of MOR is not trivial, and we have yet to successfully generate the peptide fragments required for mass spectrometry. Nevertheless, our results generated using the WT receptor, the Y336F mutant and Fyn$^{-/-}$ and MOR$^{-/-}$ mice collectively demonstrate that the pMOR$^{Y336}$ antibody is indeed detecting the Src-mediated phosphorylation of MOR at Tyr$^{336}$. We

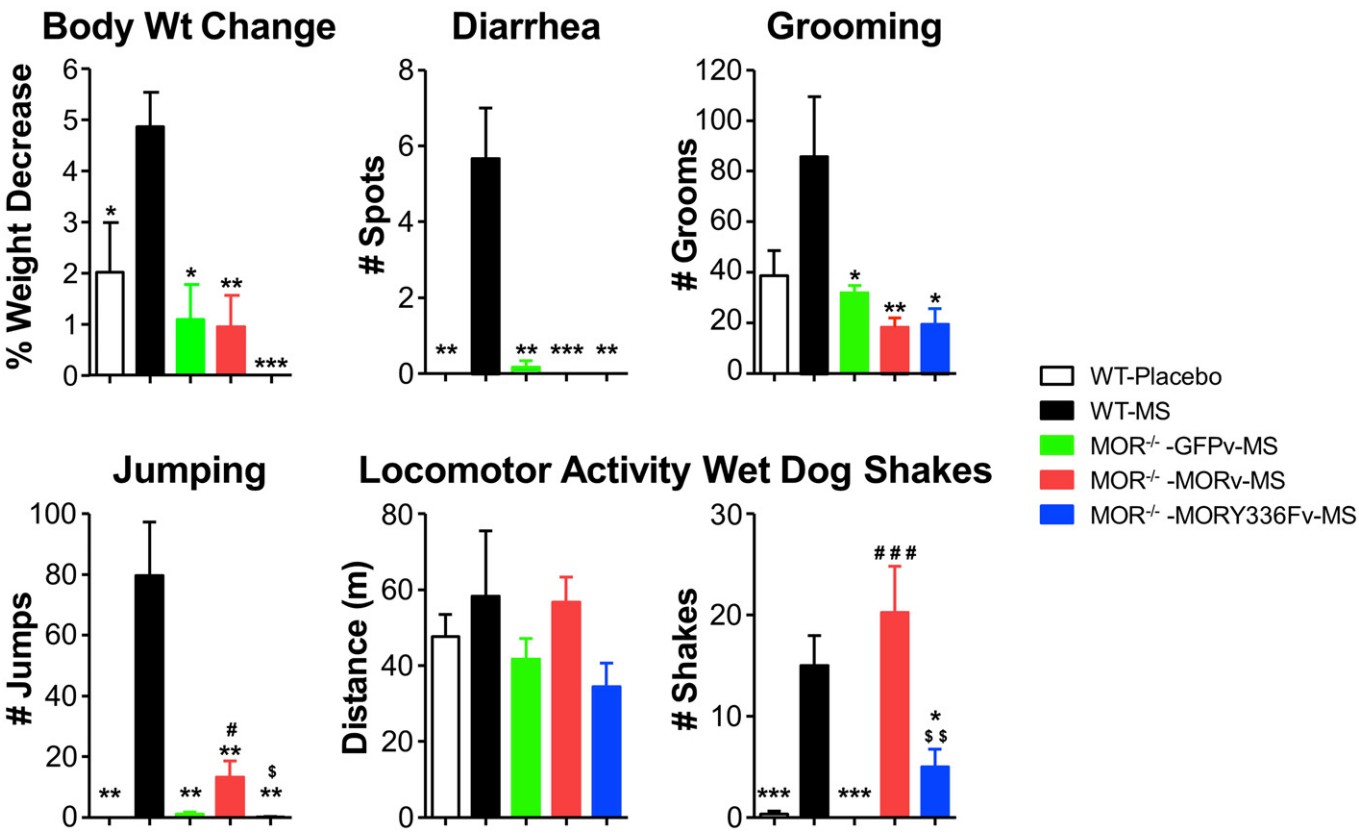

**Figure 6.   Restoration of naloxone-precipitated somatic withdrawal signs in MOR$^{-/-}$ mice with WT MOR but not MORY336F lentivirus injected into the LC.**

Construction of the lentiviruses containing the TH promoter that controls the expression of GFP (control), WT MORGFP, or MORY336F-GFP was as described in the Materials and Methods. MOR$^{-/-}$ mice were stereotaxically injected with the GFPv ($n = 6$), MORv ($n = 8$), or MORY336Fv ($n = 6$) 1 week prior to chronic treatment with morphine and the subsequent monitoring of the naloxone-precipitated somatic withdrawal signs as described. In parallel, chronic morphine treatment and naloxone-precipitated withdrawal were carried out in WT mice (WT-MS, $n = 6$). Placebo-implanted WT mice injected with naloxone do not show withdrawal (WT-Placebo, $n = 6$). The somatic withdrawal signs of body weight change, diarrhea, grooming, jumping, locomotor activity, and WDS were assessed for 30 min. Significant differences among the groups (MOR$^{-/-}$-GFPv-MS, MOR$^{-/-}$-MORv-MS, and MOR$^{-/-}$-MORY336F-MS) were determined using one-way ANOVA, followed by Duncans *post hoc* tests. ***$P < 0.001$, **$P < 0.01$ and *$P < 0.05$ relative to WT-MS; $^{###}P < 0.001$ and $^{#}P < 0.05$ significant differences between MOR$^{-/-}$-GFPv-MS and MOR$^{-/-}$-MORv-MS; $^{\$\$}P < 0.01$ and $^{\$}P < 0.05$ significant differences between MOR$^{-/-}$-MORv-MS and MOR$^{-/-}$-MORY336Fv-MS. The bars and errors represent the means ± SEM. Exact *P*-values are in Appendix Table S1.

also did not observe basal levels of pMOR$^{Y336}$ in mice implanted with a placebo pellet (Fig 2A). This result is in line with previous studies that showed that there is a basal MOR phosphorylation of Ser$^{363}$ and Thr$^{370}$, but not Tyr$^{336}$, in the absence of morphine or with naloxone alone (El Kouhen *et al*, 2001; Zhang *et al*, 2009). Western blots from the LC from mice implanted with either placebo or morphine pellets revealed basal levels of pSrc$^{Y416}$ (Fig 2A). This observation does not affect the key fact that morphine pellets implantation and subsequent naloxone treatment provoke a significant increase of pSrc$^{Y416}$ concomitant with MOR$^{Y336}$ phosphorylation. More importantly, this increase in Src$^{Y416}$ phosphorylation was totally blocked by the Src inhibitor AZD0530 (Fig 2A, lane 4).

It is of interest to note that the *in vivo* phosphorylation of MOR at Tyr$^{336}$ and the formation of the non-canonical signaling complex occur only after chronic morphine treatment and naloxone-precipitated withdrawal (Figs 1A–D and 2A–D), since we did not detect pMOR$^{Y336}$ and increases of pSrc$^{Y416}$ from the LC of mice in the presence of morphine pellet alone. However, according to previous *in vitro* results, there may be a pre-existing complex of Src and MOR

in the presence of chronic morphine treatment only. Indeed, the prolonged morphine treatment of cultured cells caused a transitory increase in pSrc that reached a maximum level after 2 h of treatment but decreased 4 h later (Zhang *et al*, 2009). When pSrc decreased, Src was still highly present in the complex in the presence of morphine. Then, the consecutive exposure to naloxone not only amplified the phosphorylation but also the quantity of Src in this pre-existing complex suggesting recruitment of the kinase (Zhang *et al*, 2009). If such an *in vivo* increase in MOR phosphorylation and signal complex formation will lead to AC superactivation as observed with *in vitro* clonal cell lines (Zhang *et al*, 2013), then the subsequent aberrant increase in PKA and CREB activities that leads to the adaptation to chronic exposure to opiates will not occur as long as agonists such as morphine remain bound to the receptor. Also, if withdrawal and the adverse motivational state are the keys to drug addiction as suggested (Koob, 2009; Koob & Volkow, 2010), then the modulation of the neural substrates and circuitry that contribute to drug craving will not occur unless the agonist is dissociated from the receptor either by naloxone competition or the

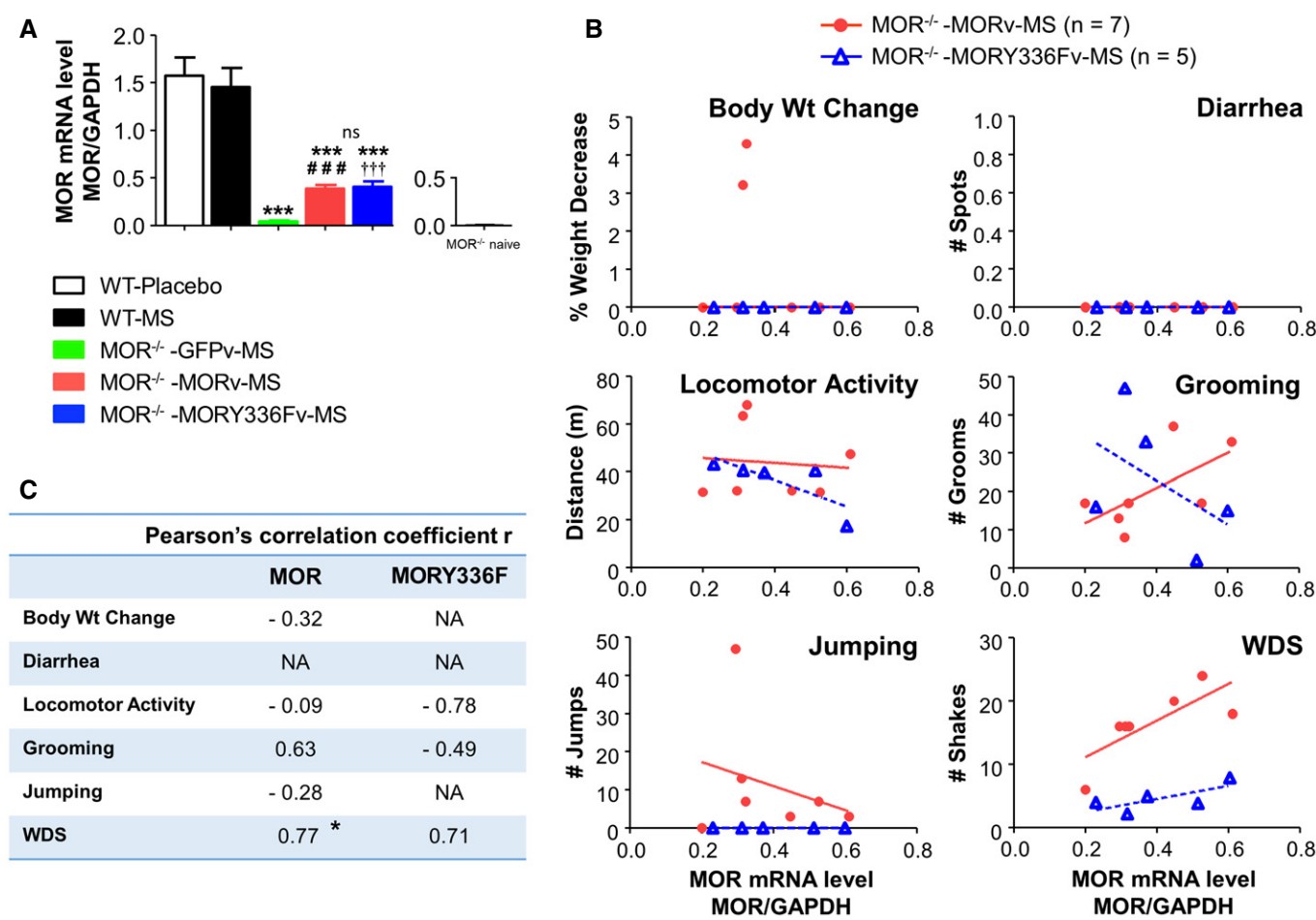

**Figure 7. Similar MOR gene expression between the WT MOR- and MORY336F-transferred MOR$^{-/-}$ mice and regression analyses.**

A   Quantitative real-time PCR (RT–qPCR) showing the level of MOR gene expression in the LC of placebo-treated WT mice (WT-Placebo, $n = 3$), and morphine-dependent mice undergoing naloxone-precipitated withdrawal such as WT (WT-MS, $n = 3$), GFP-transferred MOR$^{-/-}$ (MOR$^{-/-}$-GFPv-MS, $n = 4$), WT MORGFP-transferred MOR$^{-/-}$ (MOR$^{-/-}$-MORv-MS, $n = 7$), and MORY336F-GFP-transferred MOR$^{-/-}$ (MOR$^{-/-}$-MORY336Fv-MS, $n = 5$) mice presented in Fig 6. MOR mRNA levels are expressed relative to GAPDH mRNA levels in the LC. Each value represents the mean ± SEM of at least four independent experiments. Significant differences among the groups (MOR$^{-/-}$-GFPv-MS, MOR$^{-/-}$-MORv-MS, and MOR$^{-/-}$-MORY336Fv-MS) were determined using one-way ANOVA, followed by Duncan's *post hoc* comparison. ***$P < 0.001$ relative to WT-MS; ###$P < 0.001$ significant differences between MOR$^{-/-}$-GFPv-MS and MOR$^{-/-}$-MORv-MS; †††$P < 0.001$ significant differences between MOR$^{-/-}$-GFPv–MS and MOR$^{-/-}$-MORY336Fv-MS; ns, no significant differences between MOR$^{-/-}$-MORv-MS and MOR$^{-/-}$-MORY336Fv-MS.

B   Regression analyses of the MOR expression levels correlated with the withdrawal scores in several mice that were taken randomly from the MOR$^{-/-}$-MORv-MS and MOR$^{-/-}$-MORY336Fv-MS groups presented in Fig 6.

C   The Pearson's correlation coefficient r showed a significant and strong correlation between the level of MOR gene expression and the number of wet dog shakes (WDS) in the TH-MORGFPv-transferred MOR$^{-/-}$ mice. *$P < 0.01$, significant r in TH-MORGFPv-transferred MOR$^{-/-}$ mice.

Data information: Exact *P*-values are in Appendix Table S2.
Source data are available online for this figure.

decrease in agonist concentration that occurs in opiate abstinence. To define the transcripts that are involved in withdrawal and the adverse motivational state and therefore drug addiction, the transcripts that have roles in such a state must be distinguished from those that participate in the other states of the addiction cycle. Although the alteration of PKA or CREB activity in the NAc clearly modulates the morphine reward (Self *et al*, 1998; Barrot *et al*, 2002), the observed alterations in the downstream targets of PKA and CREB might not reflect those participating in the withdrawal/adverse motivational state due to the prolonged activation of PKA or CREB that occurs when the agonist remains bound to the receptor.

To avoid unwanted modifications of the downstream targets of PKA, CREB, or Src kinase, studying the effects of the inhibition of Src kinase activity on naloxone-precipitated withdrawal whether with the use of AZD0530, Fyn$^{-/-}$ mice, or Fyn shRNA becomes significant and coherent only when it is combined with the use of phosphorylation-deficient Y336F mutant MOR in MOR$^{-/-}$ mice. The aim is essentially to block MOR$^{Y336}$ phosphorylation. We assume that the reduction in the morphine withdrawal symptoms (i.e., jumping) when MOR$^{Y336}$ phosphorylation is blocked or inhibited by Src inhibitors or receptor mutation, prevents the formation of the complex Grb/SOS/Ras/Raf-1 with

the receptor and, thus, phosphorylation and activation of AC5/6 by Raf-1. Because Tyr$^{336}$ of MOR is itself one of the substrates of Src kinase, the alteration in opiate withdrawal signs resulting from the direct phosphorylation of Src kinase of MOR$^{Y336}$ is one possibility; whether the tyrosine phosphorylation in the NPXXY motif serves as a new docking site to directly recruit SH2/SH3 domain-containing proteins (i.e., Src kinase) to activate AC remains to be determined. However, Src kinase activity can affect indirectly the MOR$^{Y336}$ phosphorylation levels through other signaling molecules. For example, Src kinase phosphorylates phospholipase C$\gamma$ (PLC$\gamma$) and the GPCR-kinase-interacting protein-1 (GIT1) after the activation of angiotensin II type 1 receptor and epidermal growth factor receptor (Haendeler *et al*, 2003). GIT1 has been shown to be important for GPCR internalization and acts as an integrator of Src-dependent signal transduction activated by GPCRs and receptor tyrosine kinases (Haendeler *et al*, 2003).

Some existing treatments for drug relapse focus on modulating the glutamatergic activities that regulate the dopaminergic output within the NAc-VTA reward pathway. However, if the incentive-motivational theory in which drug withdrawal episodes enhance the drug's incentive value to the extent that compulsive drug-taking and drug-seeking take over the behavioral repertoire is correct (Hutcheson *et al*, 2001), then the modulation of withdrawal episodes will definitely have a pronounced effect on behavior associated with addiction. Although the use of protein kinase inhibitors such as those for PKA (Taubenfeld *et al*, 2010) and the Src kinase inhibitor PP2 (Narita *et al*, 2006) could modulate various aspects of chronic drug effects, these kinase inhibitors may also attenuate, in a maladaptive manner, a myriad of kinase actions that are unrelated to drug addiction such as synaptic plasticity, immune cells development, or cell division (Parsons & Parsons, 2004; Babus *et al*, 2011; Sen & Johnson, 2011). For example, intracerebral microinjections with PP2 significantly suppressed the morphine-induced rewarding effect and hyperlocomotion in a dose-dependent manner (Narita *et al*, 2006) and reduced the ethanol self-administration in WT animals (Wang *et al*, 2007). In humans, a genetic study correlated a mutation of the *fyn* gene with increased alcohol consumption (Schumann *et al*, 2003). A once-daily administration of SU-6656, another selective SFK markedly and dose-dependently attenuated the naloxone-precipitated withdrawal syndrome in morphine-dependent mice (Rehni & Singh, 2011). However, treatment with Src inhibitors such as Dasatinib increases the infection rate in patients with cancer, and this was suggested to occur because Dasatinib affects the immune system by reduction of neutrophil adhesion and recruitment into injured tissue (Parsons & Parsons, 2004; Zarbock, 2012). Thus, our results not only strongly suggest that the MOR non-canonical signaling pathway, particularly with the focal event of MOR$^{Y336}$ phosphorylation, may be critical for AC superactivation during the withdrawal/negative affective stage of addiction, but they also provide a new pharmacological approach for blunting opiate withdrawal episodes without altering kinase activity. If the phosphorylation of MOR at Tyr$^{336}$ residue within the NPVLY motif can be blunted with small molecules that do not inhibit protein kinases such as PKA and Src that are critical in overall memory consolidation, only the behavioral repertoire associated with drug-taking and drug-seeking should be modified. These small molecules could lead to future novel treatment paradigms for opiate addiction.

# Materials and Methods

## Animals

Homozygous Fyn knockout (Fyn$^{-/-}$) s129 mice were obtained from Jackson Laboratory (Bar Harbor, ME) and backcrossed to C57BL/6J mice until F-13 before use. WT, heterozygous, and homozygous mice from the same litter were used in the studies. The μ-opioid receptor knockout (MOR$^{-/-}$) C57BL/6J mice were generous gifts from Dr. John Pintar (Robert Wood Johnson Medical School, Piscataway, New Jersey, USA) and were crossed with C57BL/6J mice to produce the WT and MOR$^{-/-}$ mice for the experiments. The mice were housed in a temperature-controlled (21–23°C) environment with a 12-h light/dark cycle and with food and water *ad libitum*. The experiments were conducted on 2-month-old male mice (30–35 g). All the animal procedures followed the National Institutes of Health (NIH) guidelines and were approved by the University of Minnesota Institutional Animal Care (Protocol# IACUC1303A30454) and Use Committee and the Bioethics Committee of the Shanghai Institute of Materia Medica (Shanghai, China). Animals were randomly included into experimental groups according to genotyping. The experimenters were blinded for mouse treatment or transferred gene during behavior testing.

## Development and validation of rabbit polyclonal anti-pMOR$^{Y336}$

Antigen-purified polyclonal antibodies against the phosphorylated Tyr$^{336}$ of MOR (anti-pMOR$^{Y336}$) were developed by GeneTex, Inc (Irvine, CA) by immunizing rabbits with the conjugated synthetic peptide NPVLY*AFLDENC. The ability of anti-pMOR$^{Y336}$ to detect phosphorylated MOR, but not non-phosphorylated MOR, was established by carrying out Western analyses and immunofluorescence studies on HEK293 cells stably expressing either WT-HAMOR (MOR with the hemagglutinin epitope at N-terminus), HAMORY336F, HADOR, or FlagKOR. HEK293 cells expressing these receptors were treated with 1 μM morphine (WT and mutant HAMOR), 1 μM DPDPE (HADOR), or 1 μM U50, 488 (FlagKOR) for 2 h followed by treatment with or without 10 μM naloxone (HAMOR and HADOR) or with 10 μM nor-BNI (FlagKOR). Then, the cells were either transferred to ice-cold detergent buffer (50 mM Tris–HCl, pH 7.4, 150 mM NaCl, 5 mM EDTA, 10 mM NaF, 10 mM disodium pyrophosphate, 1% Nonidet P-40 [less hydrophilic than Triton X-100 and most commonly used for immunoprecipitation], 0.5% sodium deoxycholate and 0.1% SDS containing protease and phosphatase inhibitors) for homogenization and Western analysis, or the cells were fixed with 4% paraformaldehyde in 0.1 M PBS, pH 7.4, with 4% sucrose at 4°C for immunofluorescence analysis. For Western analysis, the receptor in the supernatants (16,000 × *g*, 30 min) was immunoprecipitated using protein G-agarose and mouse monoclonal anti-HA (Covance MMS-101P, 1:200) or anti-Flag (Sigma-Aldrich F-3165, 1:200) antibodies. The proteins were eluted from the beads by heating them using the SDS–PAGE treatment buffer at 65°C for 30 min and then resolved on an 8% SDS–PAGE gel. The levels of pMOR$^{Y336}$ and total epitope-tagged receptors were examined by Western blot analysis using anti-pMOR$^{Y336}$ (1 μg/ml), rabbit anti-HA (Covance PRB-101P, 1:1,000), or anti-Flag (Sigma-Aldrich F-3165, 1:1,000) antibodies with chemifluorescence performed using a Storm 860 (Molecular Dynamics). As for the immunofluorescence studies,

after fixing the cells with paraformaldehyde at 4°C for 10 min and quenching the fixation by washing with 0.1 M glycine in PBS, the cells were permeabilized by the addition of 0.25% Triton X-100 for 5 min. The levels of total and phosphorylated receptors were determined by incubating with anti-HA (Covance MMS-101P, 1:500), or anti-Flag (Sigma-Aldrich F-3165, 1:200) and anti-pMOR$^{Y336}$ (2 µg/ml) antibodies at RT for 1 h, and then incubating with Alexa Fluor 488-conjugated goat anti-mouse (Invitrogen A-10680, 1:5,000) and Alexa Fluor 594-conjugated goat anti-rabbit (Invitrogen R37117, 1:5,000) antibodies. Immunofluorescence was visualized using a CCD camera connected to a Leica DM5500 B upright microscope (Leica, Germany). Leica Application Suite (Leica, Germany) and Metamorph (Sunnyvale, CA) software were used to analyze colocalization.

### Chronic morphine treatment and naloxone-induced withdrawal signs

Mice were chronically exposed to morphine by implanting a 75-mg morphine pellet (National Institute on Drug Abuse) or a placebo pellet subcutaneously (s.c.) for 4 days. The evening before the implantation of the morphine pellet, the mice were injected with a single dose of 2.4 mg/kg of morphine sulfate (the ED$_{50}$ value for tail-flick antinociception) to ensure the survival of all the mice. On the 4th day after pellet implantation, naloxone hydrochloride (10 mg/kg) was injected intraperitoneally (i.p.) 4 h after the pellet was removed as per the method of Yano & Takemori (1977) and Seth et al (2011). The morphine analgesic tolerance previously measured with the tail-flick test (Patrick et al, 1975; Yoburn et al, 1985; Kibaly et al, 2017) is complete at 72 h after the implantation of a 75-mg morphine pellet and persists for at least 24 h after pellet removal (Patrick et al, 1975). The mice were then placed in a locomotor chamber (18″ 18″), and the withdrawal signs were tracked and recorded with the ANY-maze software (Stoelting Co., Wood Dale, IL) for the next 30 min. In the Src kinase inhibitor and Fyn shRNA experiments, parallel studies were carried out by chronically exposing the mice to morphine by pellet implantation, or by exposing the mice to progressively increasing morphine doses via s.c. injections. In the latter case, the mice were injected with either progressive doses of 10, 20, 30, 40, and 50 mg/kg of morphine hydrochloride or saline twice a day for 5 days. On day 6, the mice were injected with saline or morphine hydrochloride (60 mg/kg), and withdrawal signs were induced by the i.p. injection of naloxone hydrochloride (10 mg/kg) 4 h later and recorded. Somatic withdrawal signs such as jumping, WDS, and weight loss were graded, and signs such as diarrhea, mastication, grooming, paw tremors, ptosis, and piloerection were monitored. The locomotor activity of the mice was also recorded during the 30-min withdrawal period. The global withdrawal signs were then determined as described (Papaleo & Contarino, 2006).

### Intracerebroventricular (i.c.v.) injection of AZD0530 difumarate

The Src kinase inhibitor was injected intracerebrally into the third ventricle. The mice were anesthetized by i.p. injecting a mixture of ketamine and xylazine (final dose: 90 mg/kg ketamine and 10 mg/kg xylazine). The mice were then mounted on a stereotaxic frame (Stoelting). After exposing the skull and removing the connective tissues using the bregma as the zero coordinate, a 0.5-mm burr hole was drilled at the anteroposterior (AP) coordinate – 0.94 mm. A metal

cannula with a plug was then secured at this position with dental cement. The mice were then allowed to recover from the procedure for 1 week prior to implanting either a placebo or a morphine (75 mg) pellet s.c. On the 4th day after pellet implantation, the pellet was removed. The mice were then anesthetized with isoflurane 3.5 h after pellet removal and mounted on the stereotaxic frame. A 10-µl Hamilton syringe was filled with AZD0530 difumarate (Axon MedChem, Reston, VA). With the implanted cannula as the guide, the syringe was lowered to 2.25 mm below the skull in a 2-min period. Two microliters of the AZD0530 solution was injected with a Quintessential Stereotaxic Injector (Stoelting) in a 10-min period so as to deliver 50 µg of the inhibitor per mouse. The syringe was kept in place for an additional 5 min before it was gradually withdrawn to prevent backflow. Thirty minutes after the injection of AZD0530, the mice were i.p. injected with naloxone (10 mg/kg). The naloxone-precipitated somatic withdrawal signs were checked and recorded.

### Viral constructs

Fyn shRNA-GFP lentivirus (titer: 5.0 × 10$^6$ TU/ml) and Fyn shRNA-GFP AAV2/9 (titer: 2.09 × 10$^{12}$ v.g./ml) were purchased from Santa Cruz Biotechnology (sc-35425-V) (Dallas, TX) and OBio (AAV-830010H Y2692) (Shanghai, China), respectively. The WT MORGFP and MORY336FGFP mutant lentivirus with the tyrosine hydroxylase (TH) promoter were constructed by cloning the WT MORGFP fragments into the pCDH-CMV promoter lentiviral vector system (System Biosciences, Mountain View, CA). Site-directed mutagenesis was used to introduce a Y336F mutation into MORGFP using the primers: 5′-TTTCATCCAGGAATGCGAAAAGAACTGGATTCAGGCA GCT-3′ and 5′-AGCTGCCTGAATCCAGTTCTTTTCGCATTCCTGGAT GAAA-3′ (QuikChange II, Agilent Technologies, Santa Clara, CA). The 2.5-kb Nhe-EcoRI TH promoter fragment, provided by Dr. Uwe Maskos (Pasteur Institute, Paris, FR), were cloned into pCDH-CMV after the removal of the CMV promoter with the SnaBI/EcoRI endonucleases. Lentivirus particles were produced by the Lipofectamine-mediated transfection of HEK293T cells with four-plasmid mixtures: pLP1, pLP2, pLP/VSVG, and pCDH-TH-MORGFP or pCDH-TH-MORY336F-GFP (ratio: 3:1:1:5.5), and were purified with iodixanol gradients and tittered by flow cytometry using HT-1080 cells as described previously (Lin et al, 2012).

### Stereotaxic injection into the locus coeruleus

For the microinjections at the LC, mice were anesthetized by the i.p. injection of a mixture of ketamine and xylazine (final dose: 90 mg/kg ketamine and 10 mg/kg xylazine). After mounting the mice on the stereotaxic frame, a midsagittal incision was made to expose the bregma and lambda. With the bregma as zero, the coordinates for the LC were anteroposterior (AP) – 5.15 mm, mediolateral (ML) ± 1.20 mm, and dorsoventral (DV) – 3.62 mm (Franklin & Paxinos, 2008). For the AZD0530 difumarate injection, the mice were implanted bilaterally with a metal guide cannulae at the same coordinates 1 week prior to the initiation of chronic morphine treatment. One microliter of either AZD0530 difumarate or virus solution was microinjected through a 31-gauge Hamilton microsyringe on each side at a rate of 0.2 µl/min using the Quintessential Stereotaxic Injector (Stoelting). The syringe needle was kept in place for an additional 5 min before it was withdrawn

to prevent backflow. For the virus injection, the mice could recover for 1 week in their home cage prior to morphine exposure. For the AZD0530 difumarate injection, the Src kinase inhibitor was injected using the implanted metal cannulae as a guide 3.5 h after pellet removal or the last dose of morphine, and 30 min prior to the i.p. injection of naloxone.

## Quantifying Fyn levels in the locus coeruleus by Western analysis

Three weeks after virus infusion, the mice were euthanized by decapitation. Coronal brain sections (1 mm thick) were obtained using a mouse brain slicer (Braintree Scientific, Braintree, MA). Bilateral LC punches were pooled from two mice (four punches). The tissue was then homogenized in 100 μl of buffer containing 1% Triton, 0.1% SDS, 150 mM NaCl, 1 mM EDTA, and protease and phosphatase inhibitors cocktail. The insoluble materials were removed by centrifugation at $10,000 \times g$ for 10 min at 4°C. The protein concentration of the supernatant was determined using the BCA assay kit. Then, 20 μl of supernatant (~20 μg) was applied to the SDS–PAGE gel for Western analysis using the rabbit polyclonal antibodies against all the Src subtypes (Thermo Fisher Scientific, Waltham, MA).

## Immunohistofluorescence analysis

The mice were anesthetized with 100 mg/kg sodium pentobarbital and then perfused with Tyrode's solution (116 mM NaCl, 5.36 mM KCl, 2 mM $MgCl_2.6H_2O$, 0.406 mM $MgSO_4.7H_2O$, 1.23 mM $NaH_2PO_4$, 3 mM glucose, 26.2 mM $NaHCO_3$, pH 7.2), followed by 4% paraformaldehyde (Sigma-Aldrich) in 0.1 M phosphate buffer (75 mM $KH_2PO_4$, 85 mM $Na_2HPO_4.7H_2O$). The brains were post-fixed in 4% paraformaldehyde overnight at 4°C, immersed in 10 and 30% sucrose until they sank to the bottom and frozen in OCT (Sakura Finetek, Tokyo, Japan). Thirty-micrometer coronal tissue sections were prepared with a cryostat (Leica Microsystems, Wetzlar, Germany) and mounted on Superfrost Plus glass slides (Thermo Fisher Scientific). The sections were incubated with either rabbit anti-TH (1:1,000, for the identification of Fyn shRNA injections and WT MOR or MORY336F injections), goat anti-TH (GeneTex GTX113016, 1:1,000), mouse anti-pSrc$^{Y416}$ (Millipore clone 9A6 #05-677, 1:500), or peptide affinity-purified rabbit anti-pMOR$^{Y336}$ (1:500) overnight at 4°C. All primary antibodies were diluted in PBS, pH 7.4, 0.5% Triton X-100, 1% BSA, and 3% donkey serum (depending on the host species of the secondary antibody). After the overnight incubation with the primary antibodies, the slides were incubated with the respective anti-rabbit, anti-goat, or anti-mouse secondary antibodies (1:2,500) conjugated to either Alexa Fluor 488, Alexa Fluor 594, or Alexa Fluor 647. Immunofluorescence was visualized using a CCD camera connected to a Leica DM5500 B upright microscope (Leica, Germany). Leica Application Suite (Leica, Germany) and Metamorph (Sunnyvale, CA) software were used to perform the colocalization analysis of the images.

## Quantitative RT–PCR (RT–qPCR)

RT–qPCR was performed according to the same procedure described by Song *et al* (2007) (Appendix Supplementary Materials and Methods). The specific primer sequences for MOR1 were: forward,

## The paper explained

### Problem

Upregulation of the cAMP pathway mediated via AC superactivation (i.e., the compensatory increase in AC activity) in response to chronic opiate administration has been shown to be one important mechanism involved in physical opiate dependence and withdrawal. We recently demonstrated that AC superactivation after prolonged opiate treatment in cell models requires the recruitment and activation of a tyrosine kinase protein, Src, and subsequent phosphorylation of the MOR at Tyr$^{336}$ by Src. Thus, our hypothesis is that elimination of MOR-Tyr$^{336}$ phosphorylation will prevent the appearance of withdrawal symptoms in mice after interruption of long-term opioid treatment or addition of an opioid antagonist (e.g., naloxone).

### Results

We observed the recruitment of Src in the vicinity of MOR and an increase in phosphorylated Tyr$^{336}$ (pY336) levels during naloxone-precipitated withdrawal in mice. The intracerebroventricular or stereotaxic injection into the locus coeruleus (LC) of a Src inhibitor (AZD0530), or Src shRNA viruses, attenuated pY336 levels, several somatic withdrawal signs, and overall global withdrawal scores in WT mice. Similar results were observed in Fyn$^{-/-}$ mice that do not express Fyn, a Src kinase subtype. The stereotaxic injection of wild-type (WT) MOR, but not the phosphorylation-deficient (Y336F) MOR, lentiviruses into the LC of MOR$^{-/-}$ mice restored somatic withdrawal jumping. These observations suggest that the Src-mediated phosphorylation of MOR at Tyr$^{336}$ is a prerequisite for opiate withdrawal.

### Impact

Treatment for opiate abuse involves substituting heroin or other opiates with another agonist (methadone) or a partial agonist (buprenorphine). However, both drugs also have potential for abuse. Our findings form the basis for the development of small allosteric ligands for the μ-opioid receptor that can inhibit receptor phosphorylation without inhibiting Src kinase activity or interfering with the other cellular functions of the Src kinase. Thus, they provide an alternative approach for the future development of drugs for the treatment of opiate abuse.

5′-CATGGCCCTCTATTCTATCGTGT-3′, and reverse, 5′-CAGCGTGC TAGTGGCTAAGG-3′ (Invitrogen, Carlsbad, CA, USA). The primers for the housekeeping gene glyceraldehyde-3-phosphate dehydrogenase (GAPDH) were as follows: forward, 5′-GGTGAAGGTCGGTGT GAACG-3′, and reverse, 5′-CTCGCTCCTGGAAGATGGTG-3′ (Invitrogen, Carlsbad, CA, USA).

## Statistical analysis

The results were analyzed using GraphPad Prism 5 (GraphPad Software Inc., La Jolla, CA, USA). The data are presented as means ± SEM. Sample size was chosen because of previous experience regarding data variability in similar models. No statistical method was used to predetermine sample size. Statistical differences between values from two groups were determined using unpaired Student's *t*-tests, whereas statistical differences between three or more groups were analyzed using ANOVAs followed by Duncan's *post hoc* comparisons. A difference of $P < 0.05$ was considered statistically significant.

**Expanded View** for this article is available online.

## Acknowledgements

This work was supported by grants from the National Institutes of Health DA031442, USA (PYL, LZ, CK, CX, KYS, PWM, HHL); from the Ministry of Science and Technology of China 2013CB835100 and 2015CB553502 (JGL); from the National Natural Science Foundation of China 81130087, 81671322 (JGL), and 81401107 (YJW); from the President's International Fellowship Initiative Program of the Chinese Academy of Sciences 2011T2S29 (PYL); and from the Youth Innovation Promotion Association of the Chinese Academy of Sciences 2017334 (YJW).

## Author contributions

LZ, CK, Y-JW, CX, KYS: design, acquisition of data, analysis and interpretation of data, drafting and revising the article; PWM: acquisition of data, analysis, and interpretation of data; HHL: contribution unpublished essential data or reagents; J-GL: conception and design, analysis and interpretation of data, drafting and revising the article; P-YL: conception and design, analysis and interpretation of data, writing and revising the article.

## Conflict of interest

The authors declare that they have no conflict of interest.

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
