## [Review Process File · EMBO Molecular Medicine]

Src-dependent phosphorylation of μ -opioid receptor at Tyr³³⁶ modulates opiate withdrawal.

Lei Zhang, Cherkaouia Kibaly, Yu-Jun Wang, Chi Xu, Kyu Young Song, Patrick McGarragh, Horace H. Loh, Jing-Gen Liu & Ping-Yee Law

Corresponding author: Cherkaouia Kibaly and Ping-Yee Law, University of Minnesota

Review timeline:

Submission date:	09 November 2016
Editorial Decision:	07 December 2016
Revision received:	06 June 2017
Editorial Decision:	10 July 2017
Revision received:	23 July 2017
Accepted:	24 July 2017

Transaction Report:

Editor: Céline Carret

1st Editorial Decision

07 December 2016

Thank you for the submission of your manuscript to EMBO Molecular Medicine. We have now heard back from the three referees whom we asked to evaluate your manuscript. Although the referees find the study to be of potential interest, they also raise a number of concerns that need to be fully addressed in the next final version of your article.

You will see from the comments below that while the referees find the study of interest, they also raise overlapping concerns pertaining to missing controls, details, and explanations throughout the manuscript, experiments that are not always convincing and thereby should be redone/improved, discussion that should be developed, and rewriting conclusions to better reflect the results.

We believe that all suggested experiments and text modifications are reasonable and would improve the impact of the paper and I would therefore encourage you to address these in a major revision of your work. Please note that it is EMBO Molecular Medicine policy to allow only a single round of revision and that, as acceptance or rejection of the manuscript will depend on another round of review, your responses should be as complete as possible.

I look forward to seeing a revised form of your manuscript as soon as possible.

***** Reviewer's comments *****

Referee #2 (Comments on Novelty/Model System):

Zhang et al explore a role for Src-dependent phosphorylation of mu-opioid receptor at Tyr336 in modulating opiate withdrawal. The manuscript is well written and describes the use of a variety of tools to address a role for Src phosphorylation of Tyr336 in opiate withdrawal in vivo. The authors have previously shown a role for Src kinase in signaling by MOR in vitro. Also Narita et al had used Src inhibitor to show the role of Src in modifying morphine effects in vivo. These reasons make the studies in the manuscript medium in novelty and impact.

Referee #2 (Remarks):

Zhang et al explore a role for Src-dependent phosphorylation of mu-opioid receptor at Tyr336 in modulating opiate withdrawal. The manuscript is well written and describes the use of a variety of tools to address a role for Src phosphorylation of Tyr336 in opiate withdrawal in vivo. However, the details of individual studies presented in each of the figures are missing.

Fig 1. Including a higher magnification images of cells that show co-localization would have helped. It is not clear the data in the bar graph represents how many cells/slice/animal for each of the figure - this should be indicated in figure legend. When presenting p values, when multiple samples are compared, which two samples are being compared to derive the value should be indicated in the figure (and elsewhere when multiple comparisons are made (such as in Fig 4).

Fig 2. Does the amount of pSrcY416 detected in the immunoprecipitate represent a robust signal (>10%) or a poor signal (< 1%) as compared to the total input? Western blotting data with total input is not included. What Src subtype was immunoprecipitated i.e. was the immunoprecipitate checked by MS/MS sequencing? What is the cross-reactivity of the antibody to Fyn? What are the relative levels of Fyn compared to Src and other subtypes in locus coeruleus? This information is critical if making an argument for the role of Fyn in MOR Y336 & morphine dependence.

Another point - According to the authors, the immunoprecipitation data and the data with the colocalization of the pSrc416 and pMORY336 immunofluorescence signals 'indicat(ing) that Src is recruited to the vicinity of the surface-localized MOR' (Page 6). This is an overstatement - in order to demonstrate direct interaction, additional studies using techniques such as proximity ligation-based assays are needed. The authors are advised to tone down their claim here and elsewhere when words such as 'indicate, demonstrate, show' are used so that the data is not overinterpreted.

Fig. 3. How does inhibition of Src by AZD compare to the inhibition of Fyn? In studies with direct administration of inhibitors to LC, it would be important to compare the effect of AZD to that of Src inhibitor PP2 (a well accepted and more specific inhibitor of Src) to ensure that the behavioral effects seen are in fact through Src inhibition.

Fig 4. The behavioral effects seen with the administration of MORv virus or MOR336F virus would depend directly on the relative level of expression of these proteins. A regression analysis of the relative expression correlating with the behavior is generally used to make this point.

Finally, the discussion section appears superficial. Detailed discussion of how this study relates to reported studies using the Src inhibitor, PP2 as well as other kinase inhibitors that have been shown to affect tolerance and dependence to morphine is needed.

Referee #3 (Remarks):

The manuscript by Zhang, et al., contains fascinating phenomena and is commendable in its interrelating biochemistry, and behavior. Unfortunately, enthusiasm is markedly tarnished by many elements of over interpretation, the absence of virtually any even hypothetical mechanistic framework with which to understand the remarkable ability of naloxone in tolerant preparations to

trigger phosphorylation (activation) of Src and MOR and the lack of rigorous probing of the functional (signaling) significance of MOR phosphorylation other than to demonstrate that it is necessary for somatic morphine withdrawal.

Major perceived deficiencies:

1. Although much of the biochemical context provided for opioid withdrawal revolves around the adenylyl cyclase (AC) super activation model, the connection between pMOR336 and AC super activation is never explicitly discussed.
2. The authors propose that naloxone treatment of opioid tolerant animals triggers the recruitment of Src to MOR. However, the authors need to distinguish recruitment (translocation) of Src to MOR vs. the presence of a pre-existing complex of Src and MOR in which Src becomes activated (phosphorylated) following exposure of morphine tolerant animals to naloxone. This was not done.
3. Naloxone can precipitate withdrawal within seconds. Consonance of the temporal profiles of naloxone to trigger generation of pSrcY416 and pMORY336 within the time frame of onset of somatic withdrawal is critical for establishing if formation of pSrcY416 and pMORY336 is necessary to initiate opioid withdrawal (as is strongly suggested) vs. sustaining/amplifying somatic withdrawal. This information was not provided. Naloxone was given 4 h after pellet removal. Why not immediately after? How soon thereafter could somatic withdrawal be observed and at that time, was it possible to observe increased formation of pSrcY416 and pMORY336?
4. Authors emphasize that naloxone treatment of opioid tolerant tissue induces formation of pSrcY416 and pMORY336 in tyrosine hydroxylase expressing neurons but the resolution of IHC figures provided is insufficient to support this conclusion. Co-mingling within a defined region, which is all that can be definitively concluded from images provided, does not constitute cellular co-expression. What is the functional significance of the supposed exclusive upregulation of pSrcY416 and pMORY336 in catecholamine-containing neurons within the LC? What percent of total pSrcY416 and pMORY336 fluorescence localized to tyrosine hydroxylase positive cells (only the percent of tyrosine positive cells containing pSrcY416 and pMORY336 fluorescence was provided)? How was the immunofluorescence data quantified and normalized?
5. All of the IHC experiments require pre-adsorption controls. Conclusions of specificity based on transfected cells, where targeted protein is heavily overexpressed, is not sufficient.
6. The magnitude of analgesic tolerance produced by pellet implantation and progressive systemic injection should be stated and correlated with the magnitude of naloxone-induced increase in pSrcY416 and pMORY336.
7. It is curious that the IP from the midbrain extracts from mice implanted with a placebo pellet did not reveal the presence of either pSrcY416 or pMORY336 since these results would indicate the absence of basal levels of either phosphorylated protein. This requires some perspective. [P. 8: "As shown in Fig 2A, the IP from the midbrain extracts from mice implanted with a placebo pellet did not reveal the presence of pSrcY416 or pMORY336 (lane 1)."]
8. Since the water-soluble Src inhibitor PP2 blocks morphine-induced reward and hyper-locomotion, it is strange that PP2 was not used in the current study, in order "to avoid solvent-mediated tissue damage".
9. Given the wide spectrum of substrates for Src, current results do not support the conclusion that the observed alteration in pMORY336 levels is the direct consequence of Src kinase activity.
10. No information is provided indicating the relative selectivity of AZD0530 relative to other kinase except Abl at the icv dose used, which has not been justified. Was the 50 µg icv dose of AZD0530 the lowest that had an observable effect?
11. In order to rigorously test the inference/conclusion that opioid withdrawal is a direct consequence of pSrcY416 formation, authors should compare the reduction in Src activity following

viral Fyn shRNA_v vs AZD0530 and determine if this parallels reduction in withdrawal.

12. How does the formation of a MOR complex containing Grb/SOS/Ras/Raf-1 lead to AC super activation?

13. The claimed dose responsiveness to AZD0530 is not shown.

14. Authors have convincingly demonstrated that the anti-pMORY336 antibody distinguishes non-phosphorylated from pMORY336. However, it is still necessary to demonstrate that the pMORY336 Western signal obtained from LC is specific, that it does not result from recognition of a non-targeted protein that might not be present (or is present in a much lower concentration) in transfected HEK, particularly since the expression levels of MOR in a transfected system is likely to be considerably higher than in CNS requiring much less HEK protein to be Western blotted in order to visualize MOR. Similarly, the specificity of the pSrcY416 Western signal in LC needs to be validated.

15. There is no rationale provided for using two different solubilization techniques, one with 1% Triton x-100 and 0.1% SDS in buffer for quantifying Fyn level in Western blots, the other, with 1% NP40, 0.5% sodium deoxycholate and 0.1% SDS for the IP and Westerns to measure pMORY336, HAMOR, HADOR or Flag KOR. Why the difference?

16. Cells were directly lysed (without employing a conventional membrane preparation, so we don't know where the receptors are located. This is a major limitation in understanding data within a larger context, e.g., MOR phosphorylation has been causally associated with G protein uncoupling and MOR internalization. Do the authors envision that internalized pMORY336 signals independent of G proteins to produce somatic withdrawal?

17. In Fig. 2, the normalizing Gbeta Western signal for chronic morphine/naloxone is considerably less than for placebo/naloxone, placebo/naloxone/AZD0530, or morphine/naloxone/AZD0530. Why? Is this factored into the quantification reflected by bar graphs in B and C? The corresponding legend makes no mention of this. Authors do not address the fact that in the chronic morphine/naloxone lane (lane 5), the pSrcY416 Western signal was higher than in the placebo/pellet/naloxone lane (lane 1). Furthermore, the density of the normalizing Gbeta Western signal was considerably greater in the lane 1 than lane 5. This is not reflected in the quantification reflected in the bar graph (C). Quantification bars and depicted Western signals are not congruent.

18. In Fig 3, why is jump# so different in A vs. B (approximately 75 jumps in A vs. 28 in B? I am surprised that the magnitude of reduction following AZD0530 administered via stereotaxic injection directly into the LC is comparable to that observed following i.c.v. application since icv application would be expected to have a greater distribution in the LC than that resulting from direct LC injection.

19. I find the Discussion rather anemic. There is no attempt to integrate current formulations with others in the field.

20. The title for Fig. 4 is incorrect since mutant MOR should not restore somatic withdrawal. Furthermore, in order to fully understand Fig. 4, authors should quantify protein expression of WT MOR, MOR_v and MORY336F. Otherwise comparison of responses among groups is not valid.

21. Why does stereotaxic injection of AZD into LC enhance the formation of pMORY336 in the hippocampus? Even if the middle panels of "B" were switched, it would indicate that stereotaxic injection of AZD0539 into LC blocked withdrawal-induced generation of pMORY336, which would be opposite that claimed on P. 10 ["The stereotaxic injection of 5 µg of AZD0530 bilaterally into the LC... did not affect the signals in other brain areas such as the hippocampus (Fig S5B)."]

In sum, this a fascinating beginning but there are too many unanswered questions, convoluted explication and data over interpretation for publishing what can be a very impactful line of investigation.

Referee #4 (Comments on Novelty/Model System):

Morphine withdrawal and the clinical implications are significant. The molecular pharmacology presented is illuminating and the model systems (cells and mice) are appropriate.

Referee #4 (Remarks):

This is an interesting and informative paper, but specificity controls are missing that would enhance the study's impact and credibility.

The results in figure S1 do not establish the selectivity or specificity of this pY336 antibody. DOR and KOR do show signal. Variability in HA IP and expression make direct comparisons of the transfected cells difficult. Using HA-IP is expected to restrict background and nonspecific binding. Specificity requires demonstration that proteins other than MOR don't react with the antibody, but the HA-IP precludes that.

The authors should explain why naloxone does not block morphine induced increase in pY336-ir. And further explain why naloxone and norBNI were included in the DOR and KOR images in Fig S2. The DOR and KOR images need 'no agonist' pictures for comparison.

Figure S2 shows basal staining for MOR, was this reduced by Src inhibition or is this reactivity with unphosphorylated MOR? The images from Y336F should be included for comparison.

Fig S3 is very helpful, but incomplete. The MOR^{-/-} images are important in establishing AB specificity, but agonist treatment conditions are necessary since morphine is not MOR specific. Fig 1 should include a replicate showing MOR^{-/-} images.

The authors need to comment on the failure of naloxone to block the increase pY336-ir caused by morphine treatment. If naloxone is included in this expt to mimic the naloxone ppt withdrawal, then pretreatment with morphine, followed by naloxone needs to be compared with naloxone + morphine co-admin.

Image Figure 1 specificity is difficult to assess without the corresponding western blot and without the treated MOR^{-/-} control group. Also, Fig S1 shows acute responses, whereas Fig 1 shows sustained morphine treatment. The equivalent acute response to morphine should be included in Fig 1. Since cells show pY336-ir after acute in vitro treatment, the authors should explain why acute treatment in vivo does not also robustly increase ir.

The AZD0530 experiments are important, but specificity following in vivo dosing is always a concern. Attributing its effects to inhibition of MOR phosphorylation rather than a different substrate requires further validation. The expt using viral rescue of MOR expression in LC is very important and the control showing that Y336F does not restore withdrawal is key. But controls comparing WT, MOR^{-/-}, MOR^{-/-} + viral MOR, and MOR^{-/-} + viral MOR(Y336F) are needed. How do these different mice response to acute mu agonist treatment (do the virally injected mice respond to morphine in pERK-ir, for example?).

1st Revision - authors' response

06 June 2017

GENERAL COMMENTS

We would like to bring more clarity to the main focus of this study, which is to correlate the MOR phosphorylation at Tyr³³⁶ event to the expression of opiate withdrawal in mice. The ultimate goal is to propose that the MOR phosphorylation at Tyr³³⁶ may be a pharmaceutical target to relieve withdrawal symptoms and possibly compulsory drug seeking behaviour in opiate-dependent patients.

In acute pain conditions, opioids induce analgesia by diminishing neuronal excitability by triggering of intracellular signalling events that leads to the inhibition of adenylyl cyclase (AC) activity, activation of inwardly rectifying K⁺ current and/or inhibition of calcium conductance (Law *et al.*, 2000). When pain persists, chronic opioid exposure not only leads to a blunting of these intracellular responses

but also results in a compensatory increase in intracellular cAMP levels and AC activity in response to the excessive action of the agonist (Taylor and Fleming, 2001; Zhang *et al.*, 2009, 2013). Upon the removal of the opioid from the cellular environment or the addition of an antagonist such as naloxone, the compensatory increase in AC activity becomes particularly significant and unopposed and contributes to the activation of neurons during withdrawal (Nestler, 1997). This AC superactivation phenomenon has been postulated to be the molecular basis of drug dependence and withdrawal (Koob and Bloom, 1988). A non-canonical pathway involving MOR phosphorylation at Tyr³³⁶ (MOR^{Y336}) has been proposed to be required for AC superactivation (Zhang *et al.*, 2009, 2013). A phosphorylation-deficient mutant MOR at Tyr³³⁶, which is a residue that faces the inside of the cell and could be the potential phosphorylation target of Src kinase, did not affect the acute morphine-mediated inhibition of AC activity *in vitro* (Zhang *et al.*, 2009). However, under the chronic morphine treatment condition, the mutation reduced drastically the increase in AC activity and the intracellular cAMP concentration. Moreover, the phosphorylation of MOR^{Y336} and cSrc was demonstrated to be significant only after prolonged, but not acute (< 1h), morphine treatment (Zhang *et al.*, 2009). Since MOR phosphorylation at Tyr³³⁶ seems to be functionally important in producing AC superactivation during chronic morphine administration, we tested the effect of blocking MOR phosphorylation at Tyr³³⁶ on the behavioural expression of withdrawal in mice.

In our current study, whether the Src inhibitors are selective for one of the subtypes of the Src Family Kinase (SFK), it does not affect the fact that MOR^{Y336} phosphorylation was blocked and several withdrawal signs were significantly reduced. We do not aim to use the SFK as a therapeutic target. We did not intend to use the inhibitors in our experiments to specifically pinpoint the role of one type of Src kinase, with Fyn being the major subtype that has been shown to be involved in multiple neural functions in the brain (Ohnishi *et al.*, 2011). We used the inhibitors at the beginning of the study to establish the hypothesis that the Src tyrosine kinases play a role in morphine-dependent withdrawal. Now, to address which of the SFK members is specifically involved in the expression of opiate withdrawal in mice, we utilized more specific methods, including Fyn RNA interference and lentiviral-mediated expression of the mutant MOR receptor. The lentiviral-mediated expression of a mutant MOR in the MOR^{-/-} mice is crucial because it establishes the link between withdrawal and MOR phosphorylation at Tyr³³⁶.

SPECIFIC RESPONSE TO EACH REFEREE'S SUGGESTIONS

REFEREE #2

We thank the reviewer for her/his positive and kind general comments about the paper.

Major points:

1. *“Fig 1. Including a higher magnification images of cells that show co-localization would have helped. It is not clear the data in the bar graph represents how many cells/slice/animal for each of the figure - this should be indicated in figure legend. When presenting p values, when multiple samples are compared, which two samples are being compared to derive the value should be indicated in the figure (and elsewhere when multiple comparisons are made (such as in Fig 4).”*

In Fig 1A (now Fig 3A), we added a picture at higher magnification to show the colocalization between pSrc^{Y416} and pMOR^{Y336}. We indicated the number of animals in the legend of the Fig 3C (bar graph): 3 mice for each genotype. The % of colocalization results from the quantification of the colocalization in 3 mice/genotype, 4 slices/mouse. This information was added to the legend of Fig 3C. We also provided (page 7) the one-way ANOVA that was performed on the data from the histogram in Fig 3C ($F_{(2,19)} = 303.9, P < 0.0001$). We added to the legend of Fig 3C: “Significant differences among the groups (WT, MOR^{-/-}, and Fyn^{-/-}) were determined using one-way ANOVA, followed by Duncan’s post hoc comparison. *** $P < 0.001$ ”

relative to WT; $^{##}P < 0.01$ significant differences between MOR^{-/-} and Fyn^{-/-}.” We showed the comparison on the histogram in Fig 3C.

2. **“Fig 2.** Does the amount of pSrcY416 detected in the immunoprecipitate represent a robust signal (>10%) or a poor signal (< 1%) as compared to the total input? Western blotting data with total input is not included. What Src subtype was immunoprecipitated i.e. was the immunoprecipitate checked by MS/MS sequencing? What is the cross-reactivity of the antibody to Fyn? What are the relative levels of Fyn compared to Src and other subtypes in locus coeruleus? This information is critical if making an argument for the role of Fyn in MOR Y336 & morphine dependence.”

We initially cropped out the total input from the western blot in Fig 2A (now Fig 4A). We re-added the total input right on the top of the pMOR^{Y336} band. The amount of pSrc^{Y416} detected in the immunoprecipitate represents a robust signal (>10%) compared to the total input.

Regarding which Src subtype was immunoprecipitated, we did not check by MS/MS sequencing. At this step of the study, we do not want to restrict our correlation between the phosphorylation of MOR^{Y336} and the naloxone-precipitated withdrawal state to the activation of only one Src kinase in particular. Moreover, Fyn is the major subtype that has been shown to be involved in multiple neural functions in the brain (Ohnishi *et al.*, 2011), as mentioned in the manuscript (page 8, line 11). To specifically address the role of phosphorylated Fyn in the morphine-dependent mice, the immunoprecipitates from the midbrain extracts could be detected with a monoclonal antibody specifically directed against Fyn. However, because Fyn is known to be similar in amino acid sequence to Hck, Lck, Yes1, Src, and Lyn, there would be cross-reactivity with other members of the Src family. These are the reasons why, we explored the role of Fyn in morphine dependence by using small interfering RNA (siRNA) and Fyn^{-/-} mice, which overcome the limitation of the cross-reactive antibodies.

“Another point - According to the authors, the immunoprecipitation data and the data with the colocalization of the pSrc416 and pMORY336 immunofluorescence signals 'indicat(ing) that Src is recruited to the vicinity of the surface-localized MOR' (Page 6). This is an overstatement - in order to demonstrate direct interaction, additional studies using techniques such as proximity ligation-based assays are needed. The authors are advised to tone down their claim here and elsewhere when words such as 'indicate, demonstrate, show' are used so that the data is not overinterpreted.”

We agree with the suggestion to reduce the tone of our claim and use words such as “suggest and may” instead of “indicate and demonstrate ...”.

3. **“Fig. 3.** How does inhibition of Src by AZD compare to the inhibition of Fyn? In studies with direct administration of inhibitors to LC, it would be important to compare the effect of AZD to that of Src inhibitor PP2 (a well-accepted and more specific inhibitor of Src) to ensure that the behavioral effects seen are in fact through Src inhibition.”

We mentioned in the manuscript that PP2 is only soluble in organic solvents such as DMSO (water-insoluble in the text) and added that the stereotaxic injection of the inhibitor PP2, which is dissolvable only in organic solvents such as DMSO, damaged the LC (page 10, line 5). We decided to utilize AZD0530 because it is not only soluble in water but it is also one of the four SFK inhibitors (including Dasatinib, Bosutinib (SKI-606), and KX2-391) that are currently undergoing clinical evaluation in oncology (Puls *et al.*, 2011).

4. **“Fig 4.** The behavioral effects seen with the administration of MORv virus or MOR336F virus would depend directly on the relative level of expression of these proteins. A regression

analysis of the relative expression correlating with the behavior is generally used to make this point.”

We performed quantitative real-time PCR analysis (RT-qPCR) and measured the relative level of MOR gene expression in several mice that were taken randomly from the 5 groups presented in the Fig 4 (now Fig 8). The RT-qPCR analyses showed similar MOR gene expression levels between the TH-MORGFPv- and TH-Y336FGFPv-transferred MOR^{-/-} mice (newly added Fig 9A). Regression analyses of the MOR mRNA levels correlated with the morphine withdrawal scores were included in panel B of Fig 9. In Fig 9C, the Pearson’s correlation coefficient *r* showed a significant and strong correlation between the level of MOR gene expression and the number of wet dog shakes in the TH-MORGFPv-transferred MOR^{-/-} mice. However, as mentioned on manuscript page 14 (line 20), the sample is not sufficiently large (5 and 7 mice) to detect precisely the relationship between the wild-type or mutant MOR mRNA levels and the corresponding scores of each withdrawal signs.

“Finally, the discussion section appears superficial. Detailed discussion of how this study relates to reported studies using the Src inhibitor, PP2 as well as other kinase inhibitors that have been shown to affect tolerance and dependence to morphine is needed.”

We incorporated in the discussion on pages 18-19 studies that use Src kinase inhibitors such as PP2, SU-6656 or Dasatinib to attenuate morphine-induced dependence in mice as well as the maladaptive side-effects caused by treatments with the same inhibitors.

REFEREE #3

We thank the reviewer for her/his positive and kind general comments about the paper.

Major perceived deficiencies:

1. *“Although much of the biochemical context provided for opioid withdrawal revolves around the adenylyl cyclase (AC) super activation model, the connection between pMOR336 and AC super activation is never explicitly discussed.”*

In the introduction on page 5 (from line 9), we provided a description of the mechanisms by which the phosphorylation of MOR^{Y336} affects AC superactivation as established by Zhang *et al.*, 2013 (A Novel Non-canonical Signaling Pathway for the mu-Opioid Receptor in Mol Pharm).

2. *“The authors propose that naloxone treatment of opioid tolerant animals triggers the recruitment of Src to MOR. However, the authors need to distinguish recruitment (translocation) of Src to MOR vs. the presence of a pre-existing complex of Src and MOR in which Src becomes activated (phosphorylated) following exposure of morphine tolerant animals to naloxone. This was not done.”*

In our previous study (Zhang *et al.*, 2009), the MOR receptor complex within lipid rafts was immunoprecipitated (IP), and both the amount of Tyr⁴¹⁶-phosphorylated Src (pSrc) and the amount of total cSrc associated with the complex were determined in the IP. There was a time-dependent increase in the quantity of cSrc associated with MOR with a parallel increase in Src activity (pSrc) after treatment with only 1 μM morphine for 4 h. This increase in the quantity and phosphorylation of Src associated with MOR during chronic treatment was furthermore significantly amplified when naloxone was added to displace the morphine from the receptor (Fig 3, A and C from Zhang *et al.*, 2009). Phosphorylated Src and cSrc were not detected in the immunoprecipitated MOR complex in the absence of morphine in HEKMT cells or with naloxone alone. These *in vitro* results strongly suggest that there is a pre-existing complex of

Src and MOR in the presence of chronic morphine treatment and that the consecutive exposure to naloxone not only increases phosphorylation but also the quantity of Src in this pre-existing complex, thus suggesting more recruitment of the kinase (Zhang *et al.*, 2009). These results were discussed on page 16 (2nd paragraph).

3. *“Naloxone can precipitate withdrawal within seconds. Consonance of the temporal profiles of naloxone to trigger generation of pSrcY416 and pMORY336 within the time frame of onset of somatic withdrawal is critical for establishing if formation of pSrcY416 and pMORY336 is necessary to initiate opioid withdrawal (as is strongly suggested) vs. sustaining/amplifying somatic withdrawal. This information was not provided. Naloxone was given 4 h after pellet removal. Why not immediately after? How soon thereafter could somatic withdrawal be observed and at that time, was it possible to observe increased formation of pSrcY416 and pMORY336?”*

Indeed, somatic withdrawal signs, such as jumping and wet dog shaking, can be observed within 1-2 minutes after naloxone injection. That is why we recorded the withdrawal signs as soon as naloxone was injected. This was performed for the next 30 min, after which the withdrawal signs generally disappear. Whether the formation of pSrc^{Y416} and pMOR^{Y336} is necessary to initiate opioid withdrawal vs. sustaining/amplifying is a difficult question to address because on a cellular level, we assume that each molecules of naloxone would not bind to each receptor at the same time and, consequently, that somatic withdrawal signs would not be expressed at the same time. Additionally, in term of sustaining withdrawal, does it mean increasing the time during which the signs are expressed from 30 min to more? That is not the case in our study because the withdrawal scores decrease when the phosphorylation of Src^{Y416} and MOR^{Y336} is blocked.

As mentioned in the discussion, “the subsequent aberrant increase in PKA and CREB activities that leads to the adaptation to chronic exposure to opiates will not occur as long as agonists such as morphine remain bound to the receptor. The modulation of the neural substrates and circuitry that contribute to drug craving will not occur unless the agonist is dissociated from the receptor either by naloxone competition or the decrease in agonist concentration that occurs in opiate abstinence.” We want to use the optimal conditions (i.e., decline of bound morphine + displacement with naloxone) so we can record the maximum amount of withdrawal signs possible. If morphine pellets were removed at 72 h, brain morphine declined to control levels within 6 h (Patrick *et al.*, 1975), and the frequency of jumping precipitated by naloxone appeared to be greater from 4 h (Seth *et al.*, 2011) to 8 h after morphine pellet removal (Yano and Takemori, 1977). Additionally, it was previously reported that repeated morphine treatment alone (Morphine–Vehicle) did not alter the brain reward thresholds measured at 4 h post-morphine, whereas naloxone given 4 h post-morphine resulted in a significant dose-dependent increase in the brain reward thresholds (Liu and Schulteis, 2004). For these reasons, we decided to allow the concentration of morphine to decline for 4 h before precipitating with an injection of naloxone. In the methods section on page 21 (line 15), we added references to the methods of Yano and Takemori, 1977, and Seth *et al.*, 2011.

4. *“Authors emphasize that naloxone treatment of opioid tolerant tissue induces formation of pSrcY416 and pMORY336 in tyrosine hydroxylase expressing neurons but the resolution of IHC figures provided is insufficient to support this conclusion. Co-mingling within a defined region, which is all that can be definitively concluded from images provided, does not constitute cellular co-expression. What is the functional significance of the supposed exclusive upregulation of pSrcY416 and pMORY336 in catecholamine-containing neurons within the LC? What percent of total pSrcY416 and pMORY336 fluorescence localized to tyrosine hydroxylase positive cells (only the percent of tyrosine positive cells containing pSrcY416 and pMORY336 fluorescence was provided)? How was the immunofluorescence data quantified and normalized?”*

We added a photomicrograph at a higher magnification in Fig 3A (which was previously Fig 1A) to show that there is colocalization among pSrc^{Y416}, pMOR^{Y336} and TH on a cellular level.

The results from previous studies suggest that it is primarily the locus coeruleus (LC) that plays an important role in the precipitation of the physical signs of opiate withdrawal, mainly through the expression of its motor component such as jumping, rearing and locomotor activity (Maldonado *et al.*, 1992; Maldonado and Koob, 1993; Punch *et al.*, 1997). The periaqueductal gray matter comes second. That is the reason why we focused our study of the regulation of pSrc^{Y416} and pMOR^{Y336} in the LC. Because TH-expressing neurons are the markers of the LC and because naloxone-precipitated somatic opiate withdrawal depends primarily on this structure, it was important to examine the modulation of pSrc^{Y416} and pMOR^{Y336}, specifically in the LC. However, we never stated that pSrc^{Y416} and pMOR^{Y336} upregulation is exclusive/limited to the LC. In the results on page 7 (beginning of the 2nd paragraph), we added this clarification that the LC “was demonstrated to be the primary anatomical site responsible for the expression of the motor components of opiate withdrawal such as jumping, rearing and locomotor activity in studies using electrolytic lesions, PKA inhibitors, or PKA activators (Maldonado *et al.*, 1992; Maldonado and Koob, 1993; Punch *et al.*, 1997).”

5. *“All of the IHC experiments require pre-adsorption controls. Conclusions of specificity based on transfected cells, where targeted protein is heavily overexpressed, is not sufficient.”*

We added in Fig 3B (which was previously Fig 1), the required pre-adsorption controls for the pMOR^{Y336} and pSrc^{Y416} immunofluorescence. The immunoreactivity disappeared when the pMOR^{Y336} and pSrc^{Y416} antibodies were pre-incubated with the immunoprecipitated MOR complex that had been extracted from the LC of WT mice with naloxone-precipitated withdrawal. This was mentioned at the beginning of page 7. Importantly, pMOR^{Y336} antibody did not show any immunofluorescence in MOR^{-/-} mice.

6. *“The magnitude of analgesic tolerance produced by pellet implantation and progressive systemic injection should be stated and correlated with the magnitude of naloxone-induced increase in pSrcY416 and pMORY336.”*

The time-course of the development of morphine analgesic tolerance induced by 75 mg pellet implantation does not vary substantially throughout the literature (Patrick *et al.*, 1975; Yoburn *et al.*, 1985; Kibaly *et al.*, 2017). S.c. implantation of morphine pellets induces significant analgesia and appreciable morphine brain levels as early as 20 to 30 min after implantation (Patrick *et al.*, 1975). At 1 h and 4 h, there is maximum tail-flick activity and significant increases in the concentration of morphine in the brain over this period. At 24 h after implantation, the brain morphine level is still at its peak, but tolerance to the analgesic effect begins to develop (Patrick *et al.*, 1975; Kibaly *et al.*, 2017). Although the brain level of morphine declines over the next 48 h, the decrease in tail-flick latency is more pronounced, and tolerance to tail-flick activity is complete from 36 to 72 h after implantation (Patrick *et al.*, 1975; Yoburn *et al.*, 1985; Kibaly *et al.*, 2017). If the pellets are removed at 72 h, significant tolerance persists for at least 24 h after pellet removal (Patrick *et al.*, 1975).

We added in the Materials and Methods on page 21 (line 16) that the morphine analgesic tolerance previously measured with the tail-flick test (Patrick *et al.*, 1975; Yoburn *et al.*, 1985; Kibaly *et al.*, 2017) is complete at 72 h after the implantation of a 75-mg morphine pellet and persists for at least 24 h after pellet removal (Patrick *et al.*, 1975). Thus, morphine analgesic tolerance is already present when withdrawal is precipitated with naloxone. We already noted in the results on page 7, that we did not detect pMOR^{Y336} and increases of pSrc^{Y416} from the LC of mice in the presence of 3-day morphine pellets alone or in mice implanted with placebo pellets and subjected to naloxone treatment. The *in vivo* increase in pSrc^{Y416} and pMOR^{Y336} was observed to occur only when naloxone is injected.

7. “It is curious that the IP from the midbrain extracts from mice implanted with a placebo pellet did not reveal the presence of either pSrc^{Y416} or pMOR^{Y336} since these results would indicate the absence of basal levels of either phosphorylated protein. This requires some perspective. [P. 8: “As shown in Fig 2A, the IP from the midbrain extracts from mice implanted with a placebo pellet did not reveal the presence of pSrc^{Y416} or pMOR^{Y336} (lane 1).]”

A basal MOR phosphorylation of Ser³⁶³ and Thr³⁷⁰, but not Tyr³³⁶, was previously shown (El Kouhen *et al.*, 2001). Our data on pMOR^{Y336} are also in line with an *in vitro* study (Zhang *et al.*, 2009), in which no basal levels of pMOR^{Y336} were detected in the immunoprecipitated MOR complex in the absence of morphine or with naloxone alone in HEKMT cells. Regarding pSrc^{Y416} in our current *in vivo* work, we initially thought that there was no basal pSrc^{Y416} in mice implanted with placebo. We repeated the western blot with a more sensitive equipment, and found on the updated western blots that there is a basal phosphorylation of Src^{Y416} in placebo mice (Fig 4A). Whether there is a basal level of pSrc^{Y416} in placebo mice, it does not change the key information which is that after morphine pellets implantation and naloxone treatment, there is a significant increase of pSrc^{Y416}.

On page 9 (previously page 8), we updated the sentence: “As shown in Fig 4A, the IP from the midbrain extracts from mice implanted with a placebo pellet revealed the presence of basal pSrc^{Y416} but not of pMOR^{Y336} (lane 1).”

We added to the discussion on page 16 (line 4) that “Western blots from the LC from mice implanted with either placebo or morphine pellets, revealed basal levels of pSrc^{Y416} (Fig 4). This observation does not affect the key fact that morphine pellets implantation and subsequent naloxone treatment provoke a significant increase of pSrc^{Y416} concomitant with MOR^{Y336} phosphorylation. More importantly, this increase of Src^{Y416} phosphorylation was totally blocked by the Src inhibitor AZD0530 (Fig 4A, lane 4).”

8. “Since the water-soluble Src inhibitor PP2 blocks morphine-induced reward and hyperlocomotion, it is strange that PP2 was not used in the current study, in order “to avoid solvent-mediated tissue damage”.”

We mentioned in the manuscript that PP2 is only soluble in organic solvents such as DMSO (water-insoluble in the text) and added that the stereotaxic injection of the inhibitor PP2, which is dissolvable only in organic solvents such as DMSO, damaged the LC (page 10, line 5). We decided to utilize AZD0530 because it is not only soluble in water but it is also one of the four SFK inhibitors (including Dasatinib, Bosutinib (SKI-606), and KX2-391) that are currently undergoing clinical evaluation in oncology (Puls *et al.*, 2011).

9. “Given the wide spectrum of substrates for Src, current results do not support the conclusion that the observed alteration in pMOR^{Y336} levels is the direct consequence of Src kinase activity.”

We do not stipulate that the alterations in opiate withdrawal signs result from the direct phosphorylation of Src kinase of MOR^{Y336}. As we noted in the discussion on page 17 (line 19), the direct MOR^{Y336} phosphorylation by Src kinase is one possibility because Tyr³³⁶ of MOR is itself one of the substrates of Src kinase; whether the tyrosine phosphorylation in the NPXXY motif serves as a new docking site to directly recruit SH2/SH3 domain-containing proteins (i.e., Src kinase) to activate AC remains to be determined. However, Src kinase activity can indirectly affect the MOR^{Y336} phosphorylation levels through other signaling molecules (discussion pages 17-18). For example, Src kinase phosphorylates phospholipase C γ (PLC γ) and the GPCR-kinase-interacting protein-1 (GIT1) after the activation of angiotensin II type 1 receptor and epidermal growth factor receptor (Haendeler *et al.*, 2003). GIT1 has been shown to be important for GPCR internalization and acts as an integrator of Src-dependent signal transduction activated by GPCRs and receptor tyrosine kinases (Haendeler *et al.*, 2003).

10. “No information is provided indicating the relative selectivity of AZD0530 relative to other kinase except *Abl* at the icv dose used, which has not been justified. Was the 50 µg icv dose of AZD0530 the lowest that had an observable effect?”

We added two new references that report the selectivity of AZD relative to other members of the SFKs (Green *et al.*, 2005, 2009) and inserted them in the beginning of page 10 (line 10): “Moreover, AZD0530 is one of the four SFK inhibitors (including Dasatinib, Bosutinib (SKI-606), and KX2–391) that are currently undergoing clinical evaluation in oncology (Puls *et al.*, 2011) and has >250-fold selectivity for the Src family over other tyrosine kinase families (Green *et al.*, 2005, 2009).”

The 50 µg icv dose of AZD0530 was indeed the lowest that had an observable effect.

11. “In order to rigorously test the inference/conclusion that opioid withdrawal is a direct consequence of *pSrcY416* formation, authors should compare the reduction in Src activity following viral *Fyn shRNA* vs AZD0530 and determine if this parallels reduction in withdrawal.”

Our aim is to provide evidence that the phosphorylation of MOR at Tyr³³⁶ may be the key event that leads to the expression of morphine-dependent withdrawal. It is not the focus of our study to establish whether the phosphorylation/activation of Src kinase at Tyr⁴¹⁶ has a direct or indirect action on morphine-dependent withdrawal signs. Src kinase plays a role, whether direct or indirect, in mediating pMOR^{Y336}-dependent withdrawal. As stated in the discussion at the end of page 18, “our results strongly suggest that the MOR non-canonical signaling pathway, particularly with the focal event of MOR^{Y336} phosphorylation, may be critical for AC superactivation during the withdrawal/negative affective stage of addiction.” Additionally, it is difficult to compare the amount of Fyn that is knocked-down by shRNA virus with the AZD inhibition of the enzyme activities, because Fyn like all Src kinases, needs to be phosphorylated prior to activation. Whether knocking down the enzyme level will compare similarly to the AZD inhibition of the enzymatic activity, this cannot be determined.

12. “How does the formation of a MOR complex containing *Grb/SOS/Ras/Raf-1* lead to AC super activation?”

Please see comment #1. We have added details of this non-canonical signaling pathway in the introduction on page 5 and discussion on page 17. This has previously been described in Zhang *et al.*, 2013.

13. “The claimed dose responsiveness to AZD0530 is not shown.”

We added a Supplementary Fig S1 in the Appendix that shows the dose-response of AZD0530 injected into the LC. We tested 3 different doses: 2.5 µg, 5 µg, and 10 µg. As described in the legend of the figure, the 5 µg dose of AZD0530 was the lowest that caused a consistent significant inhibition of most of the measured naloxone-precipitated withdrawal signs (body weight loss, body tremors, jumping, rearing, mastication, and piloerection). We inserted the reference to the Appendix Fig S1 into the text on page 11.

14. “Authors have convincingly demonstrated that the anti-pMORY336 antibody distinguishes non-phosphorylated from pMORY336. However, it is still necessary to demonstrate that the pMORY336 Western signal obtained from LC is specific, that it does not result from recognition of a non-targeted protein that might not be present (or is present in a much lower concentration) in transfected HEK, particularly since the expression levels of MOR in a transfected system is likely to be considerably higher than in CNS requiring much less HEK protein to be Western blotted in order to visualize MOR. Similarly, the specificity of the *pSrcY416* Western signal in LC needs to be validated.”

We added in Fig 4A (previously Fig2A) the required pre-adsorption control for the detection of pMOR^{Y336} with western blot. The band disappeared when the pMOR^{Y336} was pre-incubated with its specific phospho-peptide (NPVL(pY)AFLDENC; GeneTex). However, for the pSrc^{Y416} antibody, we only added references from the literature (page 7, line 5) because it is well-characterized and used in many reported studies, especially for western blot. Moreover, we showed residual Src activities (Src needs to be phosphorylated at Tyr⁴¹⁶ before activation) in the LC from the Fyn^{-/-} mice (Fig 4A, right). The reduced bands correlate with the sparsely detectable immunoreactivity in the photomicrographs in Fig 3C (previously Fig 1B) and Fig EV1 (previously Fig S3). This should be a clear indication of the specificity of the antibody. This has been added on the right side of Fig 4A + legend (previously Fig 2A).

15. *“There is no rationale provided for using two different solubilization techniques, one with 1% Triton x-100 and 0.1% SDS in buffer for quantifying Fyn level in Western blots, the other, with 1% NP40, 0.5% sodium deoxycholate and 0.1% SDS for the IP and Westerns to measure pMORY336, HAMOR, HADOR or Flag KOR. Why the difference?”*

They are both denaturing methods. We used the second solubilization technique containing Nonidet P-40 for the IP (followed by western blot) because Nonidet P-40 is less strong than 1% Triton-X100 at detaching protein-protein interactions, such as the IP complex, including the transfected receptor, Src kinases, protein G-agarose and mouse monoclonal anti-HA or anti-Flag antibodies. In the Materials and Methods on page 20 (line 11), we added “Nonidet P-40 [less hydrophilic than Triton X-100 and most commonly used for immunoprecipitation]”.

Note: As mentioned in the text, we used Nonidet P-40 (also called Igepal), which is not to be confused with NP40 (also called Tergitol Type NP-40). The Shell product Nonidet P-40 is an octylphenoxypolyethoxyethanol whereas the Tergitol Type NP-40 is a nonylphenoxypolyethoxyethanol. Both products are chemically different. However, Triton X-100 is an octylphenoxypolyethoxyethanol, such as Nonidet P-40. Both Nonidet P-40 and Triton X-100 products are chemically similar and closely related. They have similar properties: they are both milder non-ionic and non-denaturing agents. The only minor difference is that Nonidet P-40 is slightly less hydrophilic than Triton X-100. However, in general, Triton X-100 and Nonidet P-40 can be used interchangeably for most applications.

16. *“Cells were directly lysed (without employing a conventional membrane preparation, so we don't know where the receptors are located. This is a major limitation in understanding data within a larger context, e.g., MOR phosphorylation has been causally associated with G protein uncoupling and MOR internalization. Do the authors envision that internalized pMORY336 signals independent of G proteins to produce somatic withdrawal?”*

We do not think that it is necessary to isolate the plasma membrane because we have already reported that the phosphorylation of MOR^{Y336} by Src kinase occurs within lipid rafts in the plasma membrane (Zhao *et al.*, 2006; Zhang *et al.*, 2006). This was added to the results on page 9 (line 7-12). Precisely, AC superactivation and the ability of the Src kinase inhibitor, PP2, to attenuate morphine-induced increase of AC activity were shown to be independent from agonist-induced receptor internalization. Instead, AC superactivation requires the location of both MOR and the G_{ai2} proteins at lipid rafts (Zhao *et al.*, 2006). For example, blunting MOR internalization with the dominant-negative mutant of dynamin, K44E, did not alter the magnitude of morphine-induced AC superactivation (Zhao *et al.*, 2006). Since immunoprecipitated MOR was shown to be phosphorylated at Tyr³³⁶ by Src kinase within lipid rafts (Zhang *et al.*, 2009), the pSrc⁴¹⁶ or pMOR^{Y336} detected with our western blots are from MOR-G_{ai2}-Src signaling complexes located within lipid rafts and pulled down with the MOR N-terminus antibody.

The western blot of Fyn presented in Fig 7D (which was previously Fig 3D) was carried out to confirm the knock-down of Fyn after injection of Fyn shRNA. For this, the intracellular localization of Fyn does not matter.

17. *“In Fig. 2, the normalizing Gbeta Western signal for chronic morphine/naloxone is considerably less than for placebo/naloxone, placebo/naloxone/AZD0530, or morphine/naloxone/AZD0530. Why? Is this factored into the quantification reflected by bar graphs in B and C? The corresponding legend makes no mention of this. Authors do not address the fact that in the chronic morphine/no naloxone lane (lane 5), the pSrc^{Y416} Western signal was higher than in the placebo/pellet/naloxone lane (lane 1). Furthermore, the density of the normalizing Gbeta Western signal was considerably greater in the lane 1 than lane 5. This is not reflected in the quantification reflected in the bar graph (C). Quantification bars and depicted Western signals are not congruent.”*

Regarding the lower density of the normalizing G_β subunits in lane 3 (Morphine + Naloxone) compared to lane 1 (Placebo + Naloxone), lane 2 (Placebo + Naloxone + AZD0530), and lane 4 (Morphine + Naloxone + AZD), it is not considered significant. Similarly, there is no significant differences between the normalizing G_β subunit western signal in lane 1 (Placebo + Naloxone) compared to lane 5 (Morphine only). The densitometric quantifications of the normalizing G_β subunits from western blots of immunoprecipitated LC of 3 mice do not show any significant differences among the conditions (see new histogram Fig 4D).

In a published study, we showed that a 2-h prolonged morphine treatment alone induced an increase of pSrc^{Y416} levels in cultured cells (Figure 3C in Zhang *et al.*, 2009). The addition of naloxone after prolonged morphine treatment amplifies the increase in Src phosphorylation at Tyr³³⁶ (Fig 3C in Zhang *et al.*, 2009). That may be why there is a slight increase in pSrc^{Y416} western signal in the chronic morphine/no naloxone lane (lane 5) compared to the placebo pellet/naloxone lane (lane 1).

18. *“In Fig 3, why is jump# so different in A vs. B (approximately 75 jumps in A vs. 28 in B)? I am surprised that the magnitude of reduction following AZD0530 administered via stereotaxic injection directly into the LC is comparable to that observed following i.c.v. application since icv application would be expected to have a greater distribution in the LC than that resulting from direct LC injection.”*

Figure 3 is now Figure 7. We mentioned on page 7 (line 8) that the LC was demonstrated to be the primary site responsible for the expression of the physical signs of opiate withdrawal such as jumping, rearing and locomotor activity. The other structures such as the periaqueductal gray matter, the anterior preoptic hypothalamus, the nucleus raphe magnus, the amygdala, the nucleus accumbens, and the medial thalamus, play a weaker role in the expression of opiate withdrawal (Maldonado *et al.*, 1992; Maldonado and Koob, 1993; Punch *et al.*, 1997). Thus, it is possible that the stereotaxic injection into the LC of AZD0530 or the i.c.v. administration of the same inhibitor may induce comparable levels of reduction.

Regarding the difference in the # of jumps, there is a slight difference in the surgical procedure between the mice of Fig 7A (i.c.v. injection of AZD0530, previously Fig 3A) and the mice Fig 7B (injection of AZD0530 into the LC, previously Fig 3B). The experiments with i.c.v. injection of AZD0530 (Fig 7A) were performed with one cannula implanted into the third ventricle, whereas the injection of AZD0530 into the LC (Fig 7B) required 2 cannulae implanted bilaterally into the LC. The mice of Fig 7B required more surgery.

19. *“I find the Discussion rather anemic. There is no attempt to integrate current formulations with others in the field.”*

We incorporated into the discussion the modifications from the previous comments, such as the direct or indirect phosphorylation of MOR^{Y336} by Src kinase or studies performed with the

inhibitor PP2, the absence of basal level of MOR^{Y336} phosphorylation without morphine, and others.

20. *“The title for Fig. 4 is incorrect since mutant MOR should not restore somatic withdrawal. Furthermore, in order to fully understand Fig. 4, authors should quantify protein expression of WT MOR, MORv and MORY336F. Otherwise comparison of responses among groups is not valid.”*

The title of Fig 4 (now Fig 8) has been corrected: “Restoration of naloxone-precipitated somatic withdrawal signs in MOR^{-/-} mice with WT MOR but not MORY336F lentivirus injected into the LC.” We performed quantitative real-time PCR and measured the relative level of MOR gene expression in several mice that were taken randomly from the 5 groups presented in Fig 4 (now Fig 8). MOR gene expression was similar between the TH-MORGFPv- and TH-Y336FGFPv-transferred MOR^{-/-} mice (Fig 9A).

21. *“Why does stereotaxic injection of AZD into LC enhance the formation of pMORY336 in the hippocampus? Even if the middle panels of “B” were switched, it would indicate that stereotaxic injection of AZD0539 into LC blocked withdrawal-induced generation of pMORY336, which would be opposite that claimed on P. 10 [“The stereotaxic injection of 5 µg of AZD0530 bilaterally into the LC... did not affect the signals in other brain areas such as the hippocampus (Fig S5B).”]”*

A mistake was made with Fig S5B (now Fig 6B), which does not correspond to the original images. We have replaced Fig S5B (Fig 6B) with the correct photomicrographs. We apologize for this.

REFEREE #4

We thank the reviewer for her/his positive and kind general comments about the paper.

1. *“The results in figure S1 do not establish the selectivity or specificity of this pY336 antibody. DOR and KOR do show signal. Variability in HA IP and expression make direct comparisons of the transfected cells difficult. Using HA-IP is expected to restrict background and nonspecific binding. Specificity requires demonstration that proteins other than MOR don't react with the antibody, but the HA-IP precludes that. All of the IHC experiments require pre-adsorption controls.”*

We added in Fig 3B (which was previously Fig 1) the required pre-adsorption controls for the pMOR^{Y336} and pSrc^{Y416} immunofluorescences. The immunoreactivity disappeared when the pMOR^{Y336} and pSrc^{Y416} antibodies were pre-incubated with the immunoprecipitated MOR complex extracted from the LC of WT mice with naloxone-precipitated withdrawal. This was mentioned at the beginning of page 7. Importantly, pMOR^{Y336} antibody did not show any immunoreactivity in MOR^{-/-} mice.

2. *“The authors should explain why naloxone does not block morphine induced increase in pY336-ir. And further explain why naloxone and norBNI were included in the DOR and KOR images in Fig S2. The DOR and KOR images need 'no agonist' pictures for comparison.”*

The addition of naloxone after prolonged morphine treatment is meant to mimic the *in vivo* naloxone-precipitated withdrawal (sentence included in the legend of Fig 2 (previously Fig S2)). Naloxone is administered when the agonist has been losing its action on the receptor after prolonged agonist exposure. In this case, the antagonist more likely emphasizes the loss of the agonist's function and, thus, the intracellular modifications that are initiated during

chronic/prolonged agonist treatment. It is the cessation of morphine treatment or the dissociation of the agonist from the receptor either by naloxone competition or the decrease in agonist concentration that produces AC superactivation and withdrawal symptoms. This was added on pages 4-5. “Upon the removal of the opioid from the cellular environment or the addition of an antagonist such as naloxone, the compensatory increase of AC activity becomes particularly significant and unopposed and contributes to the activation of neurons during withdrawal (Nestler, 1997). This AC superactivation phenomenon has been postulated to be the molecular basis of drug dependence and withdrawal (Koob and Bloom, 1988).”

On the DOR and KOR images in Fig 2 (previously Fig S2), naloxone is also an antagonist of DPDPE which acts on DOR, and nor-BNI is the antagonist of U50,488 on KOR. Naloxone and nor-BNI were administered after prolonged agonist treatment to reproduce the *in vivo* naloxone-precipitated withdrawal. The anti-pMOR^{Y336} immunofluorescence in DOR and KOR cells was presented to show the specificity of the antibody. The most important point is that the pMOR^{Y336} antibody did not show any immunofluorescence in MOR^{-/-} mice. We have added the images of DOR and KOR controls.

3. “Figure S2 shows basal staining for MOR, was this reduced by Src inhibition or is this reactivity with unphosphorylated MOR? The images from Y336F should be included for comparison.”

The immunoreactivity of HA-MOR in the photos of Fig S2A (now Fig 2A, Control without morphine) corresponds more likely to unphosphorylated MOR^{Y336} since a basal MOR phosphorylation of Ser³⁶³ and Thr³⁷⁰, but not Tyr³³⁶, was shown (El Kouhen *et al.*, 2001). Our data are also in line with an *in vitro* study (Zhang *et al.*, 2009) in which no basal levels of pMOR^{Y336} were detected in the immunoprecipitated MOR complex in the absence of morphine or with naloxone alone in HEKMT cells. This information was included in the discussion on page 16 (line 2). The images of the basal staining for MOR in Y336F mutant cells were included in Fig S2B (now Fig 2B). The red color in control HEK293 is obviously dispersed in the whole cell, which is significantly different from the cells after the treatments of morphine and naloxone. The basal staining is found in all figures, but it is hidden from the much stronger signals on the membrane in other panels.

4. “Fig S3 is very helpful, but incomplete. The MOR^{-/-} images are important in establishing AB specificity, but agonist treatment conditions are necessary since morphine is not MOR specific. Fig 1 should include a replicate showing MOR^{-/-} images.”

We clarified the legend of Fig S3A and B (now Fig EV1A and B, end of page 44) that these MOR^{-/-} mice were treated with chronic morphine (pellets for 3 days) and injected with naloxone after morphine pellet removal. In Fig 3C, under the histogram showing the % of colocalized pMOR^{Y336}, pSrc^{Y416}, and TH, we included the replicates of the MOR^{-/-} and Fyn^{-/-} images of Fig S3A and B (now Fig EV1A and B).

5. “The authors need to comment on the failure of naloxone to block the increase pY336-ir caused by morphine treatment. If naloxone is included in this expt to mimic the naloxone ppt withdrawal, then pretreatment with morphine, followed by naloxone needs to be compared with naloxone + morphine co-admin.”

We explained why naloxone does not block pMOR^{Y336} in comment #2.

6. “Image Figure 1 specificity is difficult to assess without the corresponding western blot and without the treated MOR^{-/-} control group. Also, Fig S1 shows acute responses, whereas Fig 1 shows sustained morphine treatment. The equivalent acute response to morphine should be included in Fig 1. Since cells show pY336-ir after acute *in vitro* treatment, the authors should explain why acute treatment *in vivo* does not also robustly increase ir.”

In addition to the pre-adsorption controls for the pMOR^{Y336} and pSrc^{Y416} immunofluorescences in Fig 3B (which was previously Fig 1), the specificity of the anti-pMOR^{Y336} with the western blot was added on the top right of Fig 4A (previously Fig 2A). The immunoreactivity disappeared when the pMOR^{Y336} antibody was pre-incubated with its specific phospho-peptide (NPVL(pY)AFLDENC; GeneTex). The pSrc^{Y416} antibody is well-characterized and commercially available. The pSrc^{Y416} antibody is used in many reported studies, especially for western blot (please see the references added on the manuscript page 7, line 5). Additionally, we measured Src activities in the LC from the Fyn^{-/-} mice, and observed residual signals, probably due to the presence of other Src subtypes (WB included in Fig 4A). The reduced bands correlate with the sparsely detectable pSrc^{Y416} immunoreactivity in the photomicrographs of Fig 3C and Fig S3 (now Fig EV1). This should be a clear indication of the specificity of the anti-pSrc^{Y416}.

An experiment combining MOR immunoprecipitation and WB such as in Fig S1 (now Fig 1) was performed with WT mice treated with acute morphine (30 min, 10 mg/kg) and showed an absence of MOR^{Y336} and Src^{Y416} phosphorylation in the LC and hippocampus. This was added to Fig 3D.

7. “The AZD0530 experiments are important, but specificity following in vivo dosing is always a concern. Attributing its effects to inhibition of MOR phosphorylation rather than a different substrate requires further validation. The expt using viral rescue of MOR expression in LC is very important and the control showing that Y336F does not restore withdrawal is key. But controls comparing WT, MOR^{-/-}, MOR^{-/-} + viral MOR, and MOR^{-/-} + viral MOR(Y336F) are needed. How do these different mice respond to acute mu agonist treatment (do the virally injected mice respond to morphine in pERK-ir, for example?).”

We performed quantitative real-time PCR analyses and measured the relative level of MOR gene expression in several mice taken randomly from the 5 groups presented in Fig 8 (which was previously Fig 4). MOR gene expression was similar between the TH-MORGFPv- and TH-Y336FGFPv-transferred MOR^{-/-} mice as verified by RT-qPCR (Fig 9A). We have already shown in Fig 3D that there is no phosphorylation of MOR^{Y336} and Src^{Y416} in the LC and hippocampus of WT mice treated with acute morphine (30 min, 10 mg/kg) (please see comment #6). Additionally, AC superactivation is only observed in naloxone-precipitated conditions in morphine-dependent mice (introduction pages 4-5).

REFERENCES

- El Kouhen R, Burd AL, Erickson-Herbrandson LJ, Chang CY, Law PY, Loh HH (2001) Phosphorylation of Ser363, Thr370, and Ser375 residues within the carboxyl tail differentially regulates mu-opioid receptor internalization. *J Biol Chem* 276: 12774-12780
- Green T, Hennequin LF, Ple PA, Jones RJ, Clack G, Gallagher N (2005) Pre-clinical and early clinical activity of the highly selective, orally available, dual Src/Abl kinase inhibitor AZD0530. *Proc Am Assoc Cancer Res* 46 abst SY13-3
- Green TP, Fennell M, Whittaker R, Curwen J, Jacobs V, Allen J, Logie A, Hargreaves J, Hickinson DM, Wilkinson RW, Elvin P, Boyer B, Carragher N, Plé PA, Bermingham A, Holdgate GA, Ward WH, Hennequin LF, Davies BR, Costello GF (2009) Preclinical anticancer activity of the potent, oral Src inhibitor AZD0530. *Mol Oncol* 3: 248-261
- Haendeler J, Yin G, Hojo Y, Saito Y, Melaragno M, Yan C, Sharma VK, Heller M, Aebbersold R, Berk BC (2003) GIT1 mediates Src-dependent activation of phospholipase Cgamma by angiotensin II and epidermal growth factor. *J Biol Chem* 278: 49936-49944
- Kibaly C, Lin HY, Loh HH, Law PY (2017) Spinal or supraspinal phosphorylation deficiency at the MOR C-terminus does not affect morphine tolerance in vivo. *Pharmacol Res* 119: 153-168 doi: 10.1016/j.phrs.2017.01.033 [In press]
- Koob GF, Bloom FE (1988) Cellular and molecular mechanisms of drug dependence. *Science* 242: 715-723

- Law PY, Wong YH and Loh HH (2000) Molecular mechanisms and regulation of opioid receptor signaling. *Annu Rev Pharmacol Toxicol* 40: 389-430
- Liu J, Schulteis G (2004) Brain reward deficits accompany naloxone-precipitated withdrawal from acute opioid dependence. *Pharmacol Biochem Behav* 79:101-108
- Maldonado R, Stinus L, Gold LH, Koob GF (1992) Role of different brain structures in the expression of the physical morphine withdrawal syndrome. *J Pharmacol Exp Ther* 261: 669-677
- Maldonado R, Koob GF (1993) Destruction of the locus coeruleus decreases physical signs of opiate withdrawal. *Brain Res* 605: 128-138
- Nestler EJ (1997) Molecular mechanisms of opiate and cocaine addiction. *Curr Opin Neurobiol* 7: 713-719
- Ohnishi H, Murata Y, Okazawa H, Matozaki T (2011) Src family kinases: modulators of neurotransmitter receptor function and behavior. *Trends in Neurosci* 34: 629-637
- Patrick GA, Dewey WL, Spaulding TC, Harris LS (1975) Relationship of brain morphine levels to analgesic activity in acutely treated mice and rats and in pellet implanted mice. *J Pharmacol Exp Ther* 193: 876-883
- Puls LN, Eadens M, Messersmith W (2011) Current status of SRC inhibitors in solid tumor malignancies. *Oncologist* 16: 566-578
- Punch LJ, Self DW, Nestler EJ, Taylor JR (1997) Opposite modulation of opiate withdrawal behaviors on microinfusion of a protein kinase A inhibitor versus activator into the locus coeruleus or periaqueductal gray. *J Neurosci* 17: 8520-8527
- Seth V, Upadhyaya P, Moghe V, Ahmad M (2011) Role of calcium in morphine dependence and naloxone-precipitated withdrawal in mice. *J Exp Pharmacol* 3: 7-12
- Taylor DA, Fleming WW (2001) Unifying perspectives of the mechanisms underlying the development of tolerance and physical dependence to opioids. *J Pharmacol Exp Ther* 297: 11-18
- Yano I, Takemori AE (1977) Inhibition by naloxone of tolerance and dependence in mice treated acutely and chronically with morphine. *Res Commun Chem Pathol Pharmacol* 16: 721-734
- Yoburn BC, Chen J, Huang T, Inturrisi CE (1985) Pharmacokinetics and pharmacodynamics of subcutaneous morphine pellets in the rat. *J Pharmacol Exp Ther* 235: 282-286
- Zhao H, Loh HH, Law PY (2006) Adenylyl cyclase superactivation induced by long-term treatment with opioid agonist is dependent on receptor localized within lipid rafts and is independent of receptor internalization. *Mol Pharmacol* 69: 1421-1432
- Zhang L, Zhao H, Qiu Y, Loh HH, Law PY (2009) Src phosphorylation of mu-receptor is responsible for the receptor switching from an inhibitory to a stimulatory signal. *J Biol Chem* 284: 1990-2000
- Zhang L, Loh HH, Law PY (2013) A novel noncanonical signaling pathway for the mu-opioid receptor. *Mol Pharmacol* 84: 844-853

2nd Editorial Decision

10 July 2017

Thank you for the submission of your revised manuscript to EMBO Molecular Medicine and please accept my apologies for replying so late. We have now received 2 reports from the 2 referees I asked to re-review. Unfortunately, as you will see, while referee 2 is supportive, referee #3 remained unsupportive. In order to settle discrepancies, I therefore asked an editorial external advisor to help us reach a fair decision and the comments of this advisor are copied below.

After discussing within the team, and taking into account the two positive referees and advisor, we agreed that we will be able to accept your manuscript pending the following final amendments:

1) please carefully read the comments from referee #3 and reply in writing in a point-by-point response, amending the main article file as needed. Please use the comments of our advisor as guidelines for improving further the flow of the paper.

2) Following advice from our advisor, we would also encourage you to simplify the number of figures and maybe indeed moving figures 1 and 2 to Appendix would be desirable.

-Figures 2, 3, 6 and EV1: please make all scale bars similar everywhere, and white (preferably)

Please submit your revised manuscript within two weeks.

I look forward to reading a new revised version of your manuscript as soon as possible.

***** Reviewer's comments *****

Referee #2 (Remarks):

The authors have adequately addressed the majority of reviewer's concerns and hence the manuscript is acceptable for publication.

Referee #3 (Remarks):

While this manuscript has some intriguing findings, it is deficient on multiple levels. Enthusiasm is substantially diminished by the very superficial and myopic justification for what is proposed as a singular, universal theory of opioid dependence/withdrawal, lack of any attempt to integrate findings with other proposed opioid withdrawal/dependence mechanisms and a striking lack of experimental rigor, notwithstanding the wide spectrum of methodologies employed.

1. The proposition that AC superactivation singularly underlies opioid withdrawal/dependence is woefully unjustified by the authors. It ignores (1) a large amount of research subsequent to the 1988 hypothesis by Koob and Bloom, (2) the findings that while AC superactivation can be observed in the CNS, at best, its magnitude is much more modest than that observed in transfected cells, and (3) many AC isoforms actually manifest under activation following chronic morphine and its acute withdrawal -AC II, IV, and VII do not manifest superactivation, and in fact show a reduction in activity upon chronic opiate exposure, (4) phosphorylation of AC V by PKA inhibits its catalytic activity, (5) PKA phosphorylation of AC VI reduces its stimulation by G_{α} . None of these findings are included in the current formulation that the entirety of opioid dependence and withdrawal results from AC superactivation. The authors never reconcile these effects of PKA, whose activity is augmented following chronic morphine, with their statement that chronic morphine-induced augmented activity of Raf-1, phosphorylates AC V/VI, which leads to their superactivation. Thus, the intellectual premise of the study is thin at best.

2. While there is an impressively broad spectrum of methodologies used, many lack necessary rigor. (1) Just because an earlier study demonstrated that LR-located MOR was Src phosphorylated at Y336 does not mean that this exclusively occurs in LR. Furthermore, those studies were performed in HEK293 cells and mouse embryonic fibroblast cells, findings in which may not necessarily translate to LC. Thus, justification for using crude membrane fraction vs. LR fractions is very weak. (2) The use of a singular inhibitor of cSrc is not adequate, particularly since the concentration achieved following application icv or directly into brain tissue is inordinately high, i.e., since the cerebrospinal volume in a mouse is around 35 μ l (roughly 10 μ l in brain and 25 μ l in spinal cord), application of 50 μ g icv would result in 10 mM concentration. At this concentration, specificity would be very ambiguous. Similar concerns pertain to 5 μ g injected stereotaxically into LC, where the final concentration cells are exposed to may not be significantly reduced from the 50 μ g applied icv. Reliance on a single inhibitor when concentration is likely to be exceedingly raises concerns (3) The IHC is not adequate. The images don't necessarily support it being on membranes. Hi-magnification confocal, using sequential imaging, would help. The authors would need to carefully describe the filter sets they used in order to assure me that this was not simply a "bleed-through" artifact.

3. No evidence is provided that Src directly phosphorylates MOR, although that is implied throughout. The increased co-IP of activated Src with MOR following chronic exposure to morphine and naloxone challenge does not necessarily indicate that Src directly phosphorylates MOR since in their 2009 paper, authors report that there is approximately a 254% increase in association of p416Src with MOR following chronic morphine (without naloxone challenge) but authors contend that naloxone-induced withdrawal is a prerequisite for increased pY336MOR formation.

4. Authors directly infer that the incomplete blockade of somatic withdrawal signs in *Fyn*^{-/-} mice could be due to the presence of other Src isoforms, but this could also indicate the likelihood that other mechanisms mediate dependence and withdrawal. This myopia pervades this manuscript.

Along this line, the authors claim that "Although somatic withdrawal from opiates can be attenuated with the use of PKA-selective inhibitors (Punch et al, 1997), the use of such inhibitors might not be the ideal approach for the treatment of opioid addiction", without providing any justification for this claim.

5. Without characterizing LC protein expression of virally transfected TH-MORGFPv and TH-Y336FGFPv (density, affinity, G protein coupling, etc) and comparing with LC MOR in WT mice, it is not possible to properly interpret viral transfection experiments. It is not sufficient to simply show that expression of the transgenes was localized within the LC structure.

6. The MOR bands in figure 1D do not look convincing; Western blots a sub par. Why was the same anti-HA antibody used for both immunoprecipitation as well as immunoblotting to detect non-phosphorylated MOR when an N-terminally directed polyclonal antibody was available (see Fig. 4)? There is a concern that IgG could be easily detected in the immunoprecipitate when the same antibody or even an antibody from the same species is used for both immunoprecipitation and Western blotting, particularly when the signal detected is around ~55-60kDa.

7. The absence of molecular mass corresponding to the MORs detected either with anti-HA or anti-NPPLY*AFLDENC antibodies raises concerns. Do these bands agree with the predicted masses of MOR?

8. Since Fyn^{-/-} mice were obtained in s129 mice strain and back crossed to C57/BL/6J, the legitimacy of making comparisons among WT, heterozygous and homozygous mice, which would have different genetic backgrounds, is questionable.

Advisor:

1. The revised paper is very much improved: all technical issues raised by the three reviewers have been adequately addressed. I think the authors have done what is best possible for this type of in vivo work. This is why reviewer #2 is happy (and so would I be).

2. Conceptual advance: the goal of the study is clear (despite reviewer#3 criticisms), i. e. testing whether a specific signaling pathway, previously characterized in vitro (Zhang 2013), indeed operates in vivo. The answer is yes. The overall message is novel and important. The authors have now softened their conclusion, i. e. this pathway is certainly not the only contributing pathway, and the locus coeruleus is certainly not the only brain site for this mechanism. I think this is clear in the revised version, and addresses conceptual criticisms. [Authors are encouraged] to make this even clearer in their introduction.

3. Reviewer #3 second set of comments: [...] I think the authors can very easily address those in a revised text (and in fact, have already addressed most of them in their revised version).

4. One comment: I think the new Figures 1 and 2 (controls requested by the reviewers) should be [Appendix] Suppl Figures (as was the case in the original paper), as well as Figure 9 (the correlation analysis requested by reviewer#2 is only an indication, but the data may be strengthened to become a main figure). But this is only a suggestion.

So overall, I would ask the authors to modify their text further, in order to address reviewer#3 second set of comments, but would not ask anymore experiments [...].

2nd Revision - authors' response

23 July 2017

GENERAL COMMENTS

1. *“Please carefully read the comments from referee #3 and reply in writing in a point-by-point response, amending the main article file as needed. Please use the comments of our advisor as guidelines for improving further the flow of the paper.”*

We responded to the comments from Referee #3 (page 2 to 8).

2. *“Following advise from our advisor, we would also encourage you to simplify the number of figures and maybe indeed moving figures 1 and 2 to Appendix would be desirable.”*

-Figures 2, 3, 6 and EV1: please make all scale bars similar everywhere, and white (preferably)

We transferred the Figures 1 and 2 to the Appendix. The Figure 1 and 2 are now Appendix Fig S1 and Fig S2, respectively. The scale bars are now white everywhere on Figures 2 (now Fig S2), 3 (now Fig 1), 6 (now Fig 4), and EV1. We decided to leave Fig 9 (now Fig 7) as a main figure because we think that the histogram represented on Fig 9A (now Fig 7A) is important. But we agree with the advisor that the data presented on Fig 9B and C (now Fig 7B and C) could be supplemental.

3. *“The TPE could be shortened. Please see online for examples of our most recent articles.”*

The Paper Explained was shortened.

4. *“Figures call out: please make sure that all figures, tables and panels are called for in the main article. For now, it looks like figures 3D and 4D are not called out.”*

Figures 3D and 4D were already called out in discussion on page 16 (line 12): “after chronic morphine treatment and naloxone-precipitated withdrawal (Figs **3A-D** and **4A-D**)”.

5. *“Please provide exact p = values, not a range. Some people found that to keep the figures clear, providing a supplemental table with all exact p -values was preferable. You are welcome to do this if you want to.”*

We provided the exact P -values in the text and figure legends for Figures 1, 2, 3, 4 and 5. The exact P -values for Figures 6, 7, EV2, EV3, EV4, EV5, and Appendix Figure S3 were presented in supplementary tables in the Appendix. We would like to mention that in some cases, the statistical tests do not give an exact P -value but $P < 0.001$ when P is extremely close to zero.

In the manuscript, we added under the legend Figures 6, 7, EV2, EV3, EV4, EV5 the sentence: “Exact P -values are in Appendix Supplementary Table x.”

6. *“As part of the EMBO Publications transparent editorial process initiative (see our Editorial at <http://embomolmed.embopress.org/content/2/9/329>), EMBO Molecular Medicine will publish online a Review Process File (RPF) to accompany accepted manuscripts.”*

Yes, we agree on publishing the RPF file.

SPECIFIC RESPONSE TO THE COMMENTS OF THE REFEREE #3

5. *“The proposition that AC superactivation singularly underlies opioid withdrawal/dependence is woefully unjustified by the authors. It ignores (1) a large amount of research subsequent to the 1988 hypothesis by Koob and Bloom, (2) the findings that while AC superactivation can be observed in the CNS, at best, its magnitude is much more modest than that observed in transfected cells, and (3) many AC isoforms actually manifest under activation following chronic morphine and its acute withdrawal -AC II, IV, and VII do not manifest superactivation, and in fact show a reduction in activity upon chronic opiate exposure, (4) phosphorylation of AC V by PKA inhibits its catalytic activity, (5) PKA phosphorylation of AC VI reduces its stimulation by Gsa. None of these findings are included in the current formulation that the entirety of opioid dependence and withdrawal results from AC superactivation. The authors never reconcile these effects of PKA, whose activity is augmented following chronic morphine, with their statement that chronic morphine-induced augmented activity of Raf-1, phosphorylates AC V/VI, which leads to their superactivation. Thus, the intellectual premise of the study is thin at best.”*

The main focus of the article is to correlate the MOR phosphorylation at Tyr³³⁶ to the expression of opiate withdrawal in mice. We previously determined *in vitro* that AC superactivation involves a non-canonical MOR signaling pathway requiring the Src-mediated phosphorylation of MOR at Tyr³³⁶ (Zhang *et al*, 2013). ABOVE ALL, the goal of our current work is to determine whether the Src-mediated phosphorylation of MOR at Tyr³³⁶ is also required in opiate withdrawal *in vivo*.

We do not exclude the existence of other mechanisms (i.e., other than AC superactivation) underlying opioid withdrawal. Thus, we modified the introduction on page 5 (lines 8-11): “This AC superactivation phenomenon, generally referred to in the literature as the upregulation of the cAMP pathway (Nestler, 2004), has been well established **as being one of the accepted molecular mechanisms of drug dependence and withdrawal** (Koob & Bloom, 1988; Nestler, 2004).”

(1) Recent studies published after Koob & Bloom (1988) such as Zhang *et al*, 2009, 2013, were mentioned in the introduction. Other recent reports are now added in the introduction (Nestler & Aghajanian, 1997; Nestler, 2004; He *et al*, 2009; Yang *et al*, 2016).

(2) It would be surprising if the AC superactivation in the CNS paralleled those observed with *in vitro* cell models, considering that opioid receptors are distributed in a small percentage of neurons and are in presence of multiple AC subtypes in the CNS. Nevertheless, AC superactivation is still observed *in vivo* as reported in several regions of the rat brain, including the locus coeruleus (Nestler & Aghajanian, 1997; He *et al*, 2009). The activation of several AC subtypes as mentioned by the reviewer reflects the activity of AC above the control level, as it is generally accepted by the field.

(3) Opioid-induced AC superactivation is isozyme specific (this is now added in the introduction page 5, lines 14-15). The fact that AC2 and AC4 do not show superactivation does not mean that AC superactivation does not exist or the other isoforms are not involved in the AC superactivation. Among the 10 different isoforms of AC (Patel *et al*, 2001), AC1, AC5 and AC8 are the only isoforms that have been implicated in these adaptive processes related to opioid dependence and withdrawal (Pierre *et al*, 2009). That is the reason why, we cited them in the introduction on page 4 (lines 12-14): “The involvement of adenylyl cyclase (AC) in chronic morphine action was demonstrated with AC5, AC1 and AC8 knockout mice, which showed the attenuation of some somatic withdrawal signs (Kim *et al*, 2006; Zachariou *et al*, 2008).” We now clearly state on page 5 (lines 15-19) that AC1, AC5, AC6 and AC8 are superactivated after chronic morphine exposure in transfected cells (Avidor-Reiss *et al*, 1996, 1997). In mice, AC1, AC5 and AC8 are more specifically implicated in the behavioral expression of withdrawal (Kim *et al*, 2006; Zachariou *et al*, 2008).

(4, 5) The negative feedback involving the inhibition of AC5/6 by PKA was described in cardiac myocytes (Kapiloff *et al*, 2009). Such negative feedback has not been reported to occur under conditions of chronic morphine treatment, naloxone-precipitated withdrawal or opioid physical dependence. Moreover, in Zhang *et al*, 2009, among the kinase inhibitors tested, the inhibitor of PKA, KT5720, does not change the magnitude of AC superactivation after prolonged morphine exposure and naloxone addition (Figure 1A in Zhang *et al*, 2009).

Our Src/Raf-1-mediated activation of AC can lead to the downstream activation of PKA signaling pathways involving DARPP-32 or CREB leading to the phenotype of the opioid withdrawal state. Involvement of DARPP-32 and CREB in opiate withdrawal has been extensively studied by Nestler and his collaborators (See, “Chronic administration of morphine increases the levels of AC1, ACVIII, PKA catalytic (C) and regulatory (RII) subunits, and several phosphoproteins, including CREB and tyrosine hydroxylase (TH) (Nestler, 2004).”). The intellectual aspect of our study is that the opioid receptor being a GPCR can activate the Ras/Raf-1 pathway and subsequent AC/PKA signaling through the phosphorylation of the receptor at Tyr³³⁶. This has never been reported and basically is a non-canonical signaling

pathway of the receptor. So, we disagree with the statement by the reviewer that “*the intellectual premise of the study is thin at best*”.

6. “*While there is an impressively broad spectrum of methodologies used, many lack necessary rigor. (1) Just because an earlier study demonstrated that LR-located MOR was Src phosphorylated at Y336 does not mean that this exclusively occurs in LR. Furthermore, those studies were performed in HEK293 cells and mouse embryonic fibroblast cells, findings in which may not necessarily translate to LC. Thus, justification for using crude membrane fraction vs. LR fractions is very weak. (2) The use of a singular inhibitor of cSrc is not adequate, particularly since the concentration achieved following application icv or directly into brain tissue is inordinately high, i.e., since the cerebrospinal volume in a mouse is around 35 μ l (roughly 10 μ l in brain and 25 μ l in spinal cord), application of 50 μ g icv would result in 10 mM concentration. At this concentration, specificity would be very ambiguous. Similar concerns pertain to 5 μ g injected stereotaxically into LC, where the final concentration cells are exposed to may not be significantly reduced from the 50 μ g applied icv. Reliance on a single inhibitor when concentration is likely to be exceedingly raises concerns (3) The IHC is not adequate. The images don't necessarily support it being on membranes. Hi-magnification confocal, using sequential imaging, would help. The authors would need to carefully describe the filter sets they used in order to assure me that this was not simply a “bleed-through” artifact.*”

(1) MOR has been shown to reside mainly in lipid raft microdomains of plasma membranes not only in HEK293 cells but also in mouse and rat brain (Huang *et al*, 2008; Zheng *et al*, 2012), and its signaling can be either impaired or enhanced upon lipid raft disruption by cholesterol removal with methyl- β -cyclodextrin (M β CD) in different cell types (Zhao *et al*, 2006; Huang *et al*, 2007; Zheng *et al*, 2008; Levitt *et al*, 2009; Qiu *et al*, 2011). We have shown that phosphorylation of MOR at Tyr³³⁶ occurred within the lipid raft (Zhang *et al*, 2009). The lipid rafts are also enriched with a variety of signaling factors, such as G proteins, GPCRs, Src and ACs (Li *et al*, 1996; Ostrom *et al*, 2001; Navratil *et al*, 2003). Extracting cholesterol with M β CD disrupts the lipid raft microdomains and attenuates the signaling of several GPCRs including opioid receptors (Navratil *et al*, 2003, Monastyrskaya *et al*, 2005, Zhang *et al*, 2006). This strongly suggests the regulation of the function of GPCRs by cholesterol and lipid rafts (Chini & Parenti, 2004; Barnett-Norris *et al*, 2005). The concept that lipid rafts function as microdomains in plasma membranes to concentrate signaling molecules for regulated activation by related receptors has been well accepted in the field (Pike, 2003; Chini & Parenti, 2004; Cohen *et al*, 2004). This is the reason why we use crude membrane preparations in our study.

(2) Our study DO NOT rely on one method ONLY. The study is not built on the solely use of a singular cSrc inhibitor, but on the combination of several specific methods, including Fyn RNA interference, Fyn^{-/-} mice and lentiviral-mediated expression of the phosphorylation-deficient MOR. The results with the cSrc inhibitor are complementary to the results with the phosphorylation-deficient MOR lentivirus, Fyn^{-/-} mice and Fyn RNA interference in WT mice. Moreover, the question we address is whether the phosphorylation of MOR at Tyr³³⁶ is required in opioid withdrawal *in vivo*. The reduction of withdrawal signs in MOR^{-/-} mice injected with the phosphorylation-deficient MOR lentivirus provides the most direct evidence of the critical role of the phosphorylation of MOR at Tyr³³⁶ in the behavioral expression of opioid withdrawal. We know that cSrc phosphorylates directly or indirectly MOR at Tyr³³⁶ (Zhang *et al*, 2009). The experiments using Fyn RNA interference, Fyn^{-/-} mice and the cSrc inhibitor to block MOR^{Y336} phosphorylation, aim at strengthening our results with the mutant MOR lentivirus. It is the combination of the different methods we used, that validates our findings.

(3) We toned down our claim that the colocalization pSrc^{Y416}/pMOR^{Y336} is on the plasma membrane on page 8 (lines 12-15): “The colocalization of the pSrc^{Y416} and pMOR^{Y336}

immunofluorescent signals was observed at proximity of the cell surface at the CA1 and CA3 regions of the hippocampus (Fig EV1D).”

Regarding the "bleed-through" artifact”, we showed the specificity of the anti-pMOR^{Y336} and anti-pSrc^{Y416} with the required pre-adsorption controls (i.e., immunofluorescences: now Fig 1B; and pMOR^{Y336} western blot: now Fig 2A). The immunoreactivity disappeared when the pMOR^{Y336} and pSrc^{Y416} antibodies were pre-incubated with the immunoprecipitated MOR complex extracted from the LC of WT mice with naloxone-precipitated withdrawal. This was mentioned on page 7. Importantly, the antibodies did not show any immunoreactivity (or sparse immunoreactivity for pSrc^{Y416}) in MOR^{-/-} and Fyn^{-/-} mice.

7. *“No evidence is provided that Src directly phosphorylates MOR, although that is implied throughout. The increased co-IP of activated Src with MOR following chronic exposure to morphine and naloxone challenge does not necessarily indicate that Src directly phosphorylates MOR since in their 2009 paper, authors report that there is approximately a 254% increase in association of p416Src with MOR following chronic morphine (without naloxone challenge) but authors contend that naloxone-induced withdrawal is a prerequisite for increased pY336MOR formation.”*

We do not imply that Src phosphorylate directly MOR. In the discussion on page 18 (line 11-14), we previously specified: “whether the tyrosine phosphorylation in the NPXXY motif serves as a new docking site to directly recruit SH2/SH3 domain-containing proteins (i.e., Src kinase) to activate AC remains to be determined. However, Src kinase activity could affect indirectly the MOR^{Y336} phosphorylation levels through other signaling molecules.” The expression “Src-mediated phosphorylation of MOR” was often used in the manuscript. The expression implies only that Src mediates MOR phosphorylation whether it is direct or indirect. We have unpublished data showing that the peptide containing ³³²NPVLY³³⁶ sequence can be phosphorylated by the Src kinase. To avoid further confusions, we added in the introduction at the end of page 5 and beginning of page 6: “The activated Src phosphorylates MOR at Tyr³³⁶, although it remains to be demonstrated whether the phosphorylation is direct or indirect.”, and on page 6 (line 6) “the direct or indirect phosphorylation of MOR”.

8. *“Authors directly infer that the incomplete blockade of somatic withdrawal signs in Fyn^{-/-} mice could be due to the presence of other Src isoforms, but this could also indicate the likelihood that other mechanisms mediate dependence and withdrawal. This myopia pervades this manuscript. Along this line, the authors claim that “Although somatic withdrawal from opiates can be attenuated with the use of PKA-selective inhibitors (Punch et al, 1997), the use of such inhibitors might not be the ideal approach for the treatment of opioid addiction”, without providing any justification for this claim.”*

We previously said in the discussion at the end of page 18 and beginning of page 19, that PKA inhibitors have off-target effects unrelated to drug addiction.

“Although the use of protein kinase inhibitors such as those for PKA (Taubenfeld *et al*, 2010) and the Src kinase inhibitor PP2 (Narita *et al*, 2006) could modulate various aspects of chronic drug effects, these kinase inhibitors may also attenuate, in a maladaptive manner, a myriad of kinase actions that are unrelated to drug addiction such as synaptic plasticity, immune cells development or cell division (Parsons & Parsons, 2004; Babus *et al*, 2011; Sen & Johnson, 2011). For example, intracerebral microinjections with PP2 significantly suppressed the morphine-induced rewarding effect and hyperlocomotion in a dose-dependent manner (Narita *et al*, 2005) and reduced the ethanol self-administration in WT animals (Wang *et al*, 2007). In humans, a genetic study correlated a mutation of the *fyn* gene with increased alcohol consumption (Schumann *et al*, 2003). A once-daily administration of SU-6656, another selective SFK markedly and dose-dependently attenuated the naloxone-precipitated withdrawal syndrome in morphine-dependent mice (Rehni & Singh, 2008). However, treatment with Src inhibitors such as Dasatinib increases the infection rate in patients with cancer, and this was

suggested to occur because Dasatinib affects the immune system by reduction of neutrophil adhesion and recruitment into injured tissue (Parsons & Parsons, 2004; Zarbock, 2012).”

In the introduction at the end of page 4, we now mention why PKA-selective inhibitors might not be the best candidates for the treatment of opioid addiction: “Indeed, because of PKA interactions with multiple signaling pathways, PKA inhibitors attenuates a myriad of kinase actions that are unrelated to drug addiction (Parsons & Parsons, 2004; Babus *et al*, 2011; Sen & Johnson, 2011).”

9. *“Without characterizing LC protein expression of virally transfected TH-MORGFPv and TH-Y336FGFPv (density, affinity, G protein coupling, etc) and comparing with LC MOR in WT mice, it is not possible to properly interpret viral transfection experiments. It is not sufficient to simply show that expression of the transgenes was localized within the LC structure.”*

We do not only show that the expressed transgenes were localized within the LC structure. We quantified via quantitative real time RT-PCR, the mRNA levels of the transgenes in the LC taken from the MOR^{-/-} mice injected with the lentivirus and subjected to naloxone-precipitated withdrawal (Fig 6). The mRNA levels of GFPv, MORv, and MORY336Fv in MOR^{-/-} mice were compared to that of WT mice and naïve MOR^{-/-} mice (see Fig 7A).

10. *“The MOR bands in figure 1D do not look convincing; Western blots a sub par. Why was the same anti-HA antibody used for both immunoprecipitation as well as immunoblotting to detect non-phosphorylated MOR when an N-terminally directed polyclonal antibody was available (see Fig. 4)? There is a concern that IgG could be easily detected in the immunoprecipitate when the same antibody or even an antibody from the same specifies is used for both immunoprecipitation and Western blotting, particularly when the signal detected is around ~55-60kDa.”*

The catalogue number for the anti-HA used for the western blot is PRB-101P and not MMS-101P. The catalogue number was misspelled. The anti-HA antibody used for IP was the mouse monoclonal anti-HA.11 antibody (MMS-101R) from Covance, and the one used for IB was the rabbit anti-HA (PRB-101P) from Covance too. We apologize for this. We have gone through the whole document to make sure there is no more spelling errors of this kind.

Thus, the IgG band was not a big issue with our protocol for IP. A diffused wide band representing the μ -opioid receptor was always observed around the 58 kDa marker due to the fact that the receptor is highly glycosylated. Only when the receptor is completely deglycosylated, a sharp band around 45 kDa is detected. This is the norm for all GPCRs that are highly glycosylated. Theoretically, we should be able to use the polyclonal rabbit anti-MOR N-terminal sequence to detect the receptor in the western analysis. However, we were unable to do so. We only use the N-terminal antibodies to pull down the receptor for further evaluation.

11. *“The absence of molecular mass corresponding to the MORs detected either with anti-HA or anti- NPVLY*AFLDENC antibodies raises concerns. Do these bands agree with the predicted masses of MOR?”*

The molecular mass of MOR is presented on the scans in Source data. The predicted molecular mass of mouse MOR is 44-45 kDa, according to the FASTA sequence. However, a band of 50-65 kDa was always detected due to its glycosylation state, no matter which antibody was used. This fact is supported by data sheets of various anti-MOR antibodies produced by a series of manufacturers.

12. *“Since Fyn^{-/-} mice were obtained in s129 mice strain and back crossed to C57/BL/6J, the legitimacy of making comparisons among WT, heterozygous and homozygous mice, which would have different genetic backgrounds, is questionable.”*

We clearly stated in Materials and Methods that homozygous *Fyn*^{-/-} mice were backcrossed to C57BL/6J until 13 generations (F-13) before use in experiments. According to Jackson Laboratories, the homozygosity is above 92% after 13 generations of backcrossing. WT, heterozygous and homozygous mice from the same litter were used in the studies.

Figure 4 in:

http://www.informatics.jax.org/mgihome/other/homepage_IntroMouse.shtml

SPECIAL THANKS TO THE ADVISOR

We thank the advisor for her/his positive general comments about the paper.

REFERENCES

- Avidor-Reiss T, Nevo I, Levy R, Pfeuffer T, Vogel Z (1996) Chronic opioid treatment induces adenylyl cyclase V superactivation. Involvement of Gbetagamma. *J Biol Chem* 271: 21309-21315
- Avidor-Reiss T, Nevo I, Saya D, Bayewitch M, Vogel Z (1997) Opiate-induced adenylyl cyclase superactivation is isozyme-specific. *J Biol Chem* 272: 5040-5047
- Babus LW, Little EM, Keenoy KE, Minami SS, Chen E, Song JM, Caviness J, Koo SY, Pak DT, Rebeck GW, Turner RS, Hoe HS (2011) Decreased dendritic spine density and abnormal spine morphology in *Fyn* knockout mice. *Brain Res* 1415: 96-102
- Barnett-Norris J, Lynch D, Reggio PH (2005) Lipids, lipid rafts and caveolae: their importance for GPCR signaling and their centrality to the endocannabinoid system. *Life Sci* 77: 1625-1639
- Chini B, Parenti M (2004) G-protein coupled receptors in lipid rafts and caveolae: how, when and why do they go there? *J Mol Endocrinol* 32: 325-338
- Cohen AW, Hnasko R, Schubert W, Lisanti MP (2004) Role of caveolae and caveolins in health and disease. *Physiol Rev* 84: 1341-1379
- Kapiloff MS, Piggott LA, Sadana R, Li J, Heredia LA, Henson E, Efendiev R, Dessauer CW (2009) An adenylyl cyclase-mAKAPbeta signaling complex regulates cAMP levels in cardiac myocytes. *J Biol Chem* 284: 23540-23546
- Koob GF, Bloom FE (1988) Cellular and molecular mechanisms of drug dependence. *Science* 242: 715-723
- Kim KS, Lee KW, Lee KW, Im JY, Yoo JY, Kim SW, Lee JK, Nestler EJ, Han PL (2006) Adenylyl cyclase type 5 (AC5) is an essential mediator of morphine action. *Proc Natl Acad Sci USA* 103: 3908-3913
- He L, Kim JA, Whistler JL (2009) Biomarkers of morphine tolerance and dependence are prevented by morphine-induced endocytosis of a mutant μ -opioid receptor. *FASEB J* 23: 4327-4334
- Huang P, Xu W, Yoon SI, Chen C, Chong PL, Unterwald EM, Liu-Chen LY (2007) Agonist treatment did not affect association of mu opioid receptors with lipid rafts and cholesterol reduction had opposite effects on the receptor-mediated signaling in rat brain and CHO cells. *Brain Res* 1184: 46-56
- Levitt ES, Clark MJ, Jenkins PM, Martens JR, and Traynor JR (2009) Differential effect of membrane cholesterol removal on mu- and delta-opioid receptors: a parallel comparison of acute and chronic signaling to adenylyl cyclase. *J Biol Chem* 284: 22108-22122
- Li S, Couet J, Lisanti MP (1996) Src tyrosine kinases, Galpha subunits, and H-Ras share a common membrane-anchored scaffolding protein, caveolin. Caveolin binding negatively regulates the auto-activation of Src tyrosine kinases. *J Biol Chem* 271: 29182-29190

- Monastyrskaya K, Hostettler A, Buergi S, Draeger A (2005) The NK1 receptor localizes to the plasma membrane microdomains, and its activation is dependent on lipid raft integrity. *J Biol Chem* 280: 7135-7146
- Narita M, Kato H, Kasukawa A, Narita M, Suzuki M, Takeuchi T, Suzuki T (2006) Role of Src family kinase in the rewarding effect and hyperlocomotion induced by morphine. *NeuroReport* 17: 115-119
- Navratil AM, Bliss SP, Berghorn KA, Haughian JM, Farmerie TA, Graham JK, Clay CM, Roberson MS (2003) Constitutive localization of the gonadotropin-releasing hormone (GnRH) receptor to low density membrane microdomains is necessary for GnRH signaling to ERK. *J Biol Chem* 278: 31593-31602
- Nestler EJ, Aghajanian GK (1997) Molecular and cellular basis of addiction. *Science* 278: 58-63
- Nestler EJ (2004) Historical review: Molecular and cellular mechanisms of opiate and cocaine addiction. *Trends Pharmacol Sci* 25: 210-218
- Ostrom RS, Gregorian C, Drenan RM, Xiang Y, Regan JW, Insel PA (2001) Receptor number and caveolar co-localization determine receptor coupling efficiency to adenylyl cyclase. *J Biol Chem* 276: 42063-42069
- Parsons SJ, Parsons JT (2004) Src family kinases, key regulators of signal transduction. *Oncogene* 23: 7906-7909
- Patel TB, Du Z, Pierre S, Cartin L, Scholich K (2001) Molecular biological approaches to unravel adenylyl cyclase signaling and function. *Gene* 269: 13-25
- Pierre S, Eschenhagen T, Geisslinger G, Scholich K (2009) Capturing adenylyl cyclases as potential drug targets. *Nat Rev Drug Discov* 8:321-335
- Pike LJ (2003) Lipid rafts: bringing order to chaos. *J Lipid Res* 44: 655-667
- Qiu Y, Wang Y, Law PY, Chen HZ, Loh HH (2011) Cholesterol regulates micro-opioid receptor-induced beta-arrestin 2 translocation to membrane lipid rafts. *Mol Pharmacol* 80: 210-218
- Rehni AK, Singh N (2011) Modulation of src-kinase attenuates naloxone-precipitated opioid withdrawal syndrome in mice. *Behav Pharmacol* 22: 182-190
- Schumann G, Rujescu D, Kissling C, Soyka M, Dahmen N, Preuss UW, Wieman S, Depner M, Wellek S, Lascorz J, Bondy B, Giegling I, Angheliescu I, Cowen MS, Poustka A, Spanagel R, Mann K, Henn FA, Szegedi A (2003) Analysis of genetic variations of protein tyrosine kinase fyn and their association with alcohol dependence in two independent cohorts. *Biol Psychiatry* 54: 1422-1426
- Sen B, Johnson FM (2011) Regulation of SRC family kinases in human cancers. *J Signal Transduct* 2011: 865819
- Taubenfeld SM, Muravieva EV, Garcia-Osta A, Alberini CM (2010) Disrupting the memory of places induced by drugs of abuse weakens motivational withdrawal in a context-dependent manner. *Proc Natl Acad Sci USA* 107: 12345-12350
- Wang J, Carnicella S, Phamluong K, Jeanblanc J, Ronesi JA, Chaudhri N, Janak PH, Lovinger DM, Ron D (2007) Ethanol induces long-term facilitation of NR2B-NMDA receptor activity in the dorsal striatum: implications for alcohol drinking behavior. *J Neurosci* 27: 3593-3602
- Yang HY, Nagpure BV, Bian JS (2016) Opioid Dependence and the Adenylyl Cyclase/cAMP Signaling. In *Neuropathology of Drug addiction and Substance Misuse Volume 3*, Preedy VR (ed) pp 449-456. King's College London, London: Academic Press
- Zachariou V, Liu R, LaPlant Q, Xiao G, Renthall W, Chan GC, Storm DR, Aghajanian G, Nestler EJ (2008) Distinct roles of adenylyl cyclase 1 and 8 in opiate dependence: Behavioral, electrophysiological, and molecular studies. *Biol Psychiatry* 63: 1013-21
- Zarbock A (2012) The shady side of dasatinib. *Blood* 119: 4817-4818
- Zhao H, Loh HH, Law PY (2006) Adenylyl cyclase superactivation induced by long-term treatment with opioid agonist is dependent on receptor localized within lipid rafts and is independent of receptor internalization. *Mol Pharmacol* 69: 1421-1432
- Zhang L, Tetrault J, Wang W, Loh HH, Law PY (2006) Short- and long-term regulation of adenylyl cyclase activity by delta-opioid receptor are mediated by Galphai2 in neuroblastoma N2A cells. *Mol Pharmacol* 69: 1810-1819
- Zhang L, Zhao H, Qiu Y, Loh HH, Law PY (2009) Src phosphorylation of mu-receptor is responsible for the receptor switching from an inhibitory to a stimulatory signal. *J Biol Chem* 284: 1990-2000
- Zhang L, Loh HH, Law PY (2013) A novel noncanonical signaling pathway for the mu-opioid receptor. *Mol Pharmacol* 84: 844-853
- Zheng H, Chu J, Qiu Y, Loh HH, and Law PY (2008) Agonist-selective signaling is determined by the receptor location within the membrane domains. *Proc Natl Acad Sci USA* 105: 9421-9426

Corresponding Author Name: Cherkaoua Kibaly
Journal Submitted to: EMBO Molecular Medicine
Manuscript Number: EMM-2016-07324